# Overcoming PINNs Failure Modes In High Dimension With Low-Rank Fourier Sum

**Natan Kaminsky** [1]   **Daniel Freedman** [2]   **Kira Radinsky** [1]

## Abstract

Physics-informed neural networks (PINNs) can be unreliable on PDEs with oscillatory, multiscale, stiff, or long-time solutions, and these difficulties worsen in high dimensions where collocation-based training yields large numerical integration error and high-variance gradients. We propose *Low-Rank Fourier Sums* (LoRFS), representing the solution as a low-rank sum of separable Fourier expansions (products of one-dimensional Fourier series across coordinates). This makes high-frequency structure explicit and enables *closed-form* evaluation of common physics-based objectives and their gradients (e.g., $L^2$ residual and variational losses), replacing sampling-based collocation estimates with analytic loss evaluation and eliminating sampling noise. We further provide theoretical results that clarify why LoRFS is particularly well suited to high-dimensional regimes. Across canonical PINN failure-mode benchmarks and their high-dimensional extensions, LoRFS consistently outperforms strong PINN baselines and remains stable in regimes where competing methods degrade.

## 1. Introduction

Partial differential equations (PDEs) underpin mathematical models across science and engineering, yet solving them becomes a central bottleneck in high effective dimension. Such regimes arise when the PDE depends on many parameters or coordinates, as in uncertainty quantification (Ghanem & Spanos, 2003), control and finance (Merton et al., 1971), and many-body physics (Carleo & Troyer, 2017). Classical discretizations scale poorly here: finite difference and finite element methods require grids or meshes whose size grows exponentially with dimension (Godunov & Bohachevsky, 1959; Huebner et al., 2001), and while spectral methods use global bases, naive tensor-product extensions still suffer exponential growth in degrees of freedom (Boyd, 2001).

Learning-based solvers have emerged as an alternative. Among them, physics-informed neural networks (PINNs) (Raissi et al., 2019; Lu et al., 2021) represent the solution with a neural network and enforce the PDE, boundary conditions, and initial conditions by minimizing a physics-based loss built from residual penalties. While effective on some problems, PINNs exhibit well-documented failure modes on challenging regimes: for oscillatory, multiscale, stiff, or long-time dynamics, optimization may stagnate or converge to an incorrect (often overly smooth) solution (Krishnapriyan et al., 2021; Rathore et al., 2024). Two factors are commonly implicated. First, standard neural networks exhibit a spectral bias (Rahaman et al., 2019): they tend to learn smooth, low-frequency components before high-frequency ones, which makes it difficult to represent oscillatory and multiscale structure within practical training budgets. Second, PINN objectives are typically enforced via collocation points; in higher dimensions, the residual norm is estimated from samples, and numerical integration error and gradient variance can overwhelm the optimization signal. In other words, the ideal integral loss is optimized through a noisy sampling-based approximation whose quality degrades rapidly with dimension. These issues are further exacerbated for PDEs with high-order operators, where repeated higher-order automatic differentiation can worsen conditioning and amplify numerical error.

In this work we introduce *Low-Rank Fourier Sums (LoRFS)*, a representation designed to directly target these bottlenecks via an explicitly spectral parameterization and analytic loss evaluation. LoRFS parameterizes the solution using a low-rank, separable Fourier-series structure: the function is expressed as a sum of rank-one terms, each a product of one-dimensional Fourier series along individual coordinates (so frequencies are represented explicitly, rather than emerging implicitly through training). This choice is motivated by two complementary properties. (i) As a global spectral basis, Fourier expansions represent oscillations and multiscale structure explicitly, rather than relying on implicit spectral

[1]Department of Computer Science, Technion - Israel Institute of Technology, Haifa, Israel [2]Department of Applied Mathematics, Tel Aviv University, Tel Aviv, Israel. Correspondence to: Natan Kaminsky <natank@campus.technion.ac.il>.

*Proceedings of the 43rd International Conference on Machine Learning*, Seoul, South Korea. PMLR 306, 2026. Copyright 2026 by the author(s).

learning during optimization where standard networks preferentially fit low frequencies. (ii) Crucially for training, the separable Fourier form makes the integrals that define common physics-based objectives (e.g., $L^2$ residual norms and variational formulations) *computable in closed form*, replacing collocation-based estimates. Together with analytic differentiation of Fourier series, LoRFS replaces collocation-based estimates with analytic evaluation of the integral objective and gradients, removing sampling-induced variance while retaining a compact parameterization.

LoRFS connects to classical spectral discretizations while mitigating their scaling limitations: relative to tensor-product Fourier expansions, its explicit low-rank structure yields a parameter count that grows roughly linearly with dimension for fixed rank and a fixed number of Fourier modes per coordinate (replacing an exponential tensor grid of coefficients with a small number of separable factors). At the same time, LoRFS aligns with the PINN training paradigm in that it solves PDEs by minimizing a physics-based objective with gradient-based optimization; the key difference is the solution representation and the analytic (rather than collocation-based) evaluation of the objective, so optimization targets the underlying continuous objective rather than a noisy sampled surrogate.

Beyond the algorithmic design, we provide a theoretical foundation for LoRFS that clarifies its favorable scaling properties. In particular, we establish approximation guarantees demonstrating that for broad classes of functions, LoRFS achieves efficient convergence rates with respect to both rank and Fourier degree, mitigating the computational bottlenecks often associated with high-dimensional approximations without incurring tensor-product growth in degrees of freedom.

Empirically, we perform an extensive evaluation across standard PDE benchmarks known to exhibit PINN failure modes, comparing LoRFS against a broad set of strong recent baselines. On the convection, reaction, and wave suites, LoRFS achieves the lowest relative errors (rMAE/rRMSE) across all settings. Compared to the strongest reported baselines, LoRFS improves by up to $\sim 800\times$ on convection (rMAE/rRMSE), $\sim 35$–$46\times$ on reaction, and $\sim 2.3$–$2.4\times$ on wave. Moreover, this advantage persists in high dimensions: as the dimension increases, several competing methods degrade sharply or become unstable, whereas LoRFS maintains consistently low relative error across the tested dimensions. This indicates robustness under the scaling regime where PINN-style solvers often fail. We release our code to enable reproducibility and facilitate follow-up work.[1]

Our contributions are threefold: (1) We propose **LoRFS**,

a low-rank separable Fourier representation with analytic (sampling-free) evaluation of physics-based integral objectives, designed to address canonical PINN failure modes. (2) We develop approximation guarantees that characterize error scaling with Fourier degree and rank, clarifying when LoRFS mitigates the scaling limitations of tensor-product spectral discretizations in high dimensions. (3) We demonstrate strong empirical performance on standard failure-mode PDE benchmarks and their high-dimensional extensions; LoRFS achieves the best rMAE/rRMSE across all cases (Table 1), with improvements ranging from $\sim 2\times$ to $\sim 800\times$ over the strongest baselines.

## 2. Related Work

**Classical Numerical Solvers.** Traditional numerical analysis relies on mesh-based techniques such as Finite Element (Huebner et al., 2001), Finite Difference (Godunov & Bohachevsky, 1959), and classical Spectral methods (Boyd, 2001) to reduce continuous PDEs to algebraic systems. While rigorous, these approaches are fundamentally limited by the curse of dimensionality: the number of grid points required to resolve a solution grows exponentially with the spatial dimension $d$. Sparse grid approximations (Bungartz & Griebel, 2004; Shen & Yu, 2010; Conrad & Marzouk, 2013) mitigate this explosion by employing hierarchical bases on pruned grids, but they typically require significant regularity and still face steep scaling costs in very high dimensions. LoRFS abandons the grid paradigm entirely, employing a continuous, mesh-free spectral representation that scales linearly with dimension for a fixed rank.

**Neural PDE Solvers and Optimization Pathologies.** Physics-Informed Neural Networks (Raissi et al., 2019) solve PDEs by parameterizing the solution with a neural network and embedding physical constraints directly into the loss function, enabling the simulation of complex systems like the Navier-Stokes equation (Jin et al., 2021). To improve upon this baseline, recent research has introduced specialized architectures such as QRes (Bu & Karpatne, 2021), PINNsFormer (Zhao et al., 2023), and KAN (Liu et al., 2024), alongside advanced training schemes like gPINN (Yu et al., 2022) and RoPINN (Wu et al., 2024). Despite these innovations, PINNs remain susceptible to severe failure modes. In these regimes, optimization often collapses toward trivial, over-smoothed solutions rather than capturing the true dynamics. While various strategies involving curriculum learning (Krishnapriyan et al., 2021), adaptive sampling (Mao & Meng, 2023), and causal training (Wang et al., 2022) have been proposed to mitigate this, they generally retain the limitations inherent to discrete point-wise evaluation. LoRFS fundamentally diverges from this approach by employing a continuous spectral parameterization

---

[1]https://github.com/nkami/lorfs

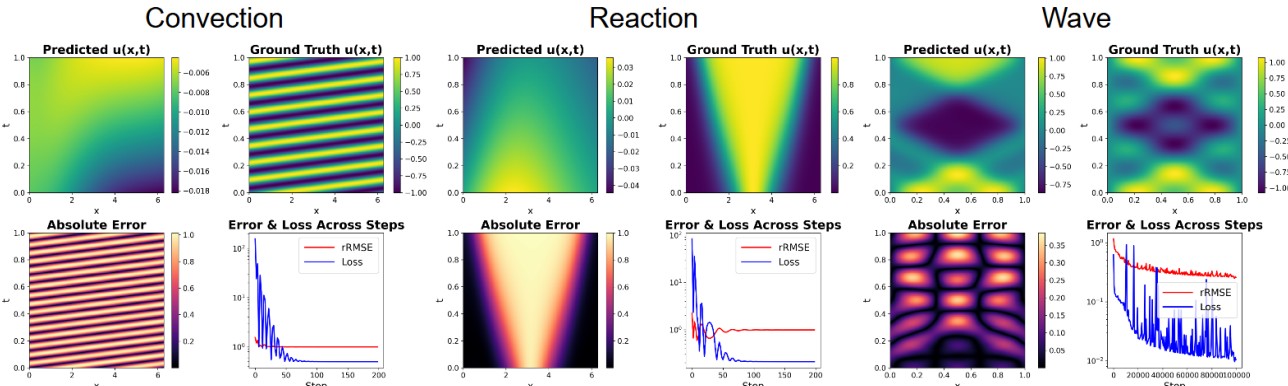

*Figure 1.* **Visualization of common failure modes in PINNs applied to the Convection, Reaction, and Wave equations.** Each column represents a distinct PDE problem. The subplots display: (top row) the PINN prediction and the ground truth solution $u(x,t)$; (bottom row) the absolute pointwise error and the evolution of the training loss versus relative RMSE (rRMSE). Despite the training loss converging to low values, the persistently high rRMSE indicates a failure to capture the correct physical dynamics, highlighting the complex loss landscapes inherent to these PDEs.

that permits exact, analytical loss computation. This eliminates the reliance on collocation points entirely, preventing the collapse into over-smooth local minima and ensuring robust convergence even for highly oscillatory problems.

**Tensor Decompositions and Separable Learning.** To reduce the computational burden of solving PDEs in high dimensions, a broad line of work exploits low-rank structure through tensor decompositions and separable parameterizations (e.g., Tensor Train and separable function learners) (Oseledets, 2011; Dektor et al., 2021; Cho et al., 2024; Wang et al., 2024b). These methods share the inductive bias of expressing solutions as sums of separable factors, which is closely related in spirit to LoRFS.

Most works in this family primarily target scalability as $d$ grows rather than explicitly addressing the optimization failure modes that arise for oscillatory, stiff, multiscale, or long-time dynamics. In contrast, LoRFS replaces neural components with differentiable Fourier series, enabling analytic differentiation via coefficient manipulation and closed-form evaluation of common physics-based losses. Consequently, while separable neural methods improve dimensional scalability, they remain vulnerable to standard optimization failure modes that LoRFS effectively overcomes.

## 3. Problem Formulation

We consider PDEs of the form

$$L[u](x) = 0, \quad x \in \Omega, \tag{1}$$

subject to boundary and/or initial conditions

$$B[u](x) = 0, \quad x \in \partial\Omega. \tag{2}$$

Here, $L$ denotes a differential operator and $B$ encodes constraints such as Dirichlet, Neumann or Robin conditions.

For time-dependent problems, we treat the time coordinate $t$ as an additional dimension within $x$, incorporating initial conditions as constraints on the corresponding boundary face. We specifically focus on rectangular domains (hyper-rectangles), defined as: $\Omega = \Omega_1 \times \cdots \times \Omega_d$. This geometry is standard in high-dimensional benchmarks, and in many engineering configurations (Binous et al., 2017; Lamelas, 2022; Pereira et al., 2018). Moreover, even when the physical domain is not naturally rectangular, box truncations are frequently used in practice—for example in computational treatments of the time-independent Schrodinger equation for many-body systems (Gerard et al., 2022; Gao & Günnemann, 2024). We assume the solution $u$ is sufficiently regular on $\Omega$.

A natural "physics loss" for enforcing the PDE and constraints is the continuous-domain least-squares objective

$$\mathcal{J}(u) = \int_\Omega \big(L[u](x)\big)^2 \, dx + \lambda \int_{\partial\Omega} \big(B[u](x)\big)^2 \, dS(x) \tag{3}$$

where $\lambda > 0$ balances interior and boundary/initial enforcement. In the PINN framework (Raissi et al., 2019; Lu et al., 2021), one parameterizes $u$ by a neural network $u_\theta$ and minimizes a Monte-Carlo approximation of (3):

$$\mathcal{L}(\theta) = \frac{1}{N} \sum_{i=1}^N \big(L[u_\theta](\tilde{x}_i)\big)^2 + \lambda \frac{1}{M} \sum_{i=1}^M \big(B[u_\theta](\hat{x}_i)\big)^2 \tag{4}$$

where $\{\tilde{x}_i\}_{i=1}^N \subset \Omega$ and $\{\hat{x}_i\}_{i=1}^M \subset \partial\Omega$ are collocation points sampled each iteration. While flexible, this discrete enforcement can require large batches for stable estimates of residual norms in high dimension and can introduce quadrature and gradient noise.

## 4. Low-Rank Fourier Sums

### 4.1. Method

To address the challenges outlined in Section 3, we introduce Low-Rank Fourier Sums. LoRFS parameterizes the solution as a low-rank sum of separable, one-dimensional complex Fourier series. Formally, we define a LoRFS $g_\theta(x)$ as:

$$g_\theta(x) = \sum_{r=1}^{s} \prod_{j=1}^{d} f_{rj}(x_j; \theta_{rj})$$
$$= f_{11}(x_1) \cdots f_{1d}(x_d) + \cdots + f_{s1}(x_1) \cdots f_{sd}(x_d).$$
(5)

Each factor $f_{rj}$ is a truncated Fourier series of degree $K$ on the coordinate $j$. For a domain $\Omega_j = [a_j, b_j]$, we express $f_{rj}$ using complex exponentials for each coordinate $x_j$:

$$f_{rj}(x_j; \theta_{rj}) = \sum_{n=-K}^{K} c_{rj,n} e^{i\omega n x_j}.$$
(6)

The learnable parameters for each factor are the complex coefficients $\theta_{rj} = \{c_{rj,n}\}_{n=-K}^{K}$. The scalar $\omega$ is a hyperparameter controlling the fundamental frequency of the basis. To ensure the global output $g_\theta(x)$ is real-valued, we enforce conjugate symmetry on the coefficients:

$$c_{rj,-n} = \overline{c_{rj,n}}.$$
(7)

This constraint implies that $c_{rj,0}$ is real and that the coefficients for $n < 0$ are determined by those for $n > 0$. Consequently, the free learnable parameters for each 1D function are the real part of the zero mode and the real and imaginary parts of the positive modes $n \in \{1, \ldots, K\}$.

The hyperparameters $s$, $K$, and $\omega$ control the rank, the spectral resolution, and the frequency scaling of the approximation, respectively. Thanks to this structure, LoRFS can be interpreted as a rank-$s$ tensor approximation of the solution in a generalized Fourier feature space. This representation is highly compact: a full tensor-product expansion would require $(2K+1)^d$ coefficients, whereas LoRFS requires only $s \cdot d \cdot (2K+1)$ parameters. This linear scaling with dimension $d$ allows LoRFS to tackle high-dimensional problems that are intractable for classical spectral methods.

The primary advantage of LoRFS is its analytic tractability under calculus operations. Since LoRFS structure is separable, $d$-dimensional integrals reduce to products of 1D integrals. Consider the integral of $g_\theta$ over the domain $\Omega = \Omega_1 \times \cdots \times \Omega_d$. Using the separability in (5), we have:

$$\int_\Omega g_\theta(x) \, dx = \sum_{r=1}^{s} \prod_{j=1}^{d} \int_{a_j}^{b_j} f_{rj}(x_j) \, dx_j.$$
(8)

Since $f_{rj}$ is a finite sum of exponentials, the 1D integral over $[a_j, b_j]$ admits a closed-form analytic solution. Substituting (6):

$$\int_{a_j}^{b_j} f_{rj}(x_j) \, dx_j = \sum_{n=-K}^{K} c_{rj,n} \int_{a_j}^{b_j} e^{i\omega n x_j} \, dx_j.$$
(9)

The integral term is explicitly solvable: it is simply $(b_j - a_j)$ if $\omega n = 0$, and $\frac{e^{i\omega n b_j} - e^{i\omega n a_j}}{i\omega n}$ otherwise. This property allows the high-dimensional integral to be computed exactly via simple summation, eliminating sampling or quadrature error. Similarly, for the $L^2$ loss computation, the integral of the square $\int_\Omega |g_\theta|^2$ involves products of exponentials which are again analytically integrable. This allows us to train using an analytic loss function rather than stochastic collocation as presented in equation (3).

### 4.2. Universal Approximation and Convergence Rates

In this section, we analyze the theoretical properties of the LoRFS architecture. We first establish that LoRFS serves as a universal approximator for continuous functions. Following this, we present our primary theoretical result: a derivation of convergence rates that remain robust against increasing dimensionality. Specifically, we identify function classes for which the approximation error decays at a rate independent of the dimension $d$. This highlights the capacity of LoRFS to effectively alleviate the curse of dimensionality, a property that, to our knowledge, has not been rigorously characterized for this specific type of high-dimensional sparse spectral approximation.

**Theorem 4.1.** *Let $u : \Omega \to \mathbb{R}$ be a continuous function defined on a compact domain $\Omega \subseteq \mathbb{T}^d$. For sufficiently large rank $s$ and frequency cutoff $K$, the Low-Rank Fourier Sum representation:*

$$g_\theta(x) = \sum_{r=1}^{s} \prod_{j=1}^{d} f_{rj}(x_j; \theta_{rj})$$
(10)

*can approximate $u(x)$ arbitrarily well in the $L^\infty$-norm. Specifically, for any $\epsilon > 0$, there exist values of $s, K$ and coefficients $\theta = \{\theta_{rj}\}$ such that:*

$$\sup_{x \in \Omega} |u(x) - g_\theta(x)| < \epsilon.$$
(11)

*Proof.* The set of all functions generated by LoRFS constitutes the algebra of multivariate trigonometric polynomials. This set is a subalgebra of $C(\mathbb{T}^d)$, as it is closed under addition and multiplication. Furthermore, this subalgebra contains the constant functions and separates points on the torus $\mathbb{T}^d$. Therefore, by the Stone–Weierstrass theorem, it is dense in $C(\mathbb{T}^d)$ with respect to the uniform norm, which implies the existence of the approximating sequence defined above. □

Despite this theoretical ability of LoRFS to approximate all continuous functions, the question remains as to whether it can do so *efficiently*. For certain function classes, such as analytic functions/Sobolev/mixed Sobolev, LoRFS exhibits favorable convergence rates. In this setting, we show that LoRFS achieves rapid convergence, with a rate that is independent of the dimension $d$. To establish this result, we restrict our attention to functions defined on the multidimensional torus $\Omega := \mathbb{T}^d$, and leverage results from spectral theory. In particular, by applying Stechkin's inequality (DeVore, 1998; Cohen et al., 2011) and proving the summability of Fourier coefficients, we demonstrate that LoRFS can mitigate the curse of dimensionality for various important function classes.

**Theorem 4.2.** *If $u$ is a member of the Sobolev space $H^m(\Omega)$, where $m > d/2$ then there exists a LoRFS representation $g_\theta$ with $s$ terms which is $\varepsilon$-close to $u$ in the sense that $\|u - g_\theta\|_{L^2} \le \varepsilon$ as long as $s = O(\varepsilon^{-2})$.*

*Proof.* See the appendix. $\square$

**Theorem 4.3.** *If $u$ is a member of the mixed smoothness Sobolev space $H^m_{\mathrm{mix}}(\Omega)$, where $m > 1/2$ then there exists a LoRFS representation $g_\theta$ with $s$ terms which is $\varepsilon$-close to $u$ in the sense that $\|u - g_\theta\|_{L^2} \le \varepsilon$ as long as $s = O(\varepsilon^{-2})$.*

*Proof.* See the appendix. $\square$

**Theorem 4.4.** *If there exists a vector $\boldsymbol{\sigma} = (\sigma_1, \sigma_2, \ldots, \sigma_d)$ with $\sigma_i > 0$ for all $i \in \{1, 2, \ldots, d\}$ such that $u$ is analytically extendable to the complex polystrip $D_{\boldsymbol{\sigma}} = \{\mathbf{z} \in \mathbb{C}^d/2\pi\mathbb{Z}^d : |\operatorname{Im} z_i| < \sigma_i, i \in \{1, 2, \ldots, d\}\}$ and $u$ is bounded in $L^2$ on the boundary hyperplanes of $D_{\boldsymbol{\sigma}}$ so the Hardy norm*

$$\|u\|^2_{H\boldsymbol{\sigma}} := \sup_{\substack{|y_i| < \sigma_i \\ 1 \le i \le d}} \int_{\mathbb{T}^d} |u(\mathbf{x} + i\mathbf{y})|^2 \, d\mathbf{x} < \infty$$

*is well-defined, then there exists a LoRFS representation $g_\theta$ with $s$ terms which is $\varepsilon$-close to $u$ in the sense that $\|u - g_\theta\|_{L^2} \le \varepsilon$ as long as $s = O(\varepsilon^{-2})$.*

*Proof.* See the appendix. $\square$

Theorems 4.2, 4.3 and 4.4 demonstrate that the asymptotic convergence rate with respect to the number of terms $s$ is independent of the dimension $d$. However, it remains to be understood how the parameter $K$, which controls the maximal frequency in LoRFS, scales with $d$ for accurate approximation. In what follows, we show that for functions in the Sobolev space $H^m(\Omega)$, where $m$ is large enough to guarantee continuity ($m > d/2$), and also for functions in the mixed smoothness Sobolev space $H^m_{\mathrm{mix}}(\Omega)$ with $m > 1/2$, LoRFS achieves polynomial convergence with respect to $K$.

**Theorem 4.5.** *If $u$ is a member of the Sobolev space $H^m(\Omega)$, where $m > d/2$ then there exists a LoRFS representation $g_\theta$ using terms up to frequency $K$ which is $\varepsilon$-close to $u$ in the sense that $\|u - g_\theta\|_{L^2} \le \varepsilon$ as long as $K = O(\varepsilon^{-\frac{1}{m}})$.*

*Proof.* See the appendix. $\square$

**Theorem 4.6.** *If $u$ is a member of the mixed smoothness Sobolev space $H^m_{\mathrm{mix}}(\Omega)$, where $m > 1/2$ then there exists a LoRFS representation $g_\theta$ using terms up to frequency $K$ which is $\varepsilon$-close to $u$ in the sense that $\|u - g_\theta\|_{L^2} \le \varepsilon$ as long as $K = O(\varepsilon^{-\frac{1}{m}})$.*

*Proof.* See the appendix. $\square$

Theorems 4.5 and 4.6 show that the required maximal frequency $K$ in each univariate component grows like $\varepsilon^{-\frac{1}{m}}$ which is smaller than $\varepsilon^{-2}$, since in both cases $m > 1/2$. This implies that for both of the function spaces $H^m(\Omega)$ and $H^m_{\mathrm{mix}}(\Omega)$ the primary complexity bottleneck in LoRFS is not the frequency cutoff $K$, but rather the number of rank-1 terms $s$ needed to achieve a given approximation error.

One would expect that the convergence of LoRFS with respect to $K$ would be superior for a smoother family of functions, for example, analytic functions, compared to what is attained for the Sobolev spaces above.

**Theorem 4.7.** *If there exists a vector $\boldsymbol{\sigma} = (\sigma_1, \sigma_2, \ldots, \sigma_d)$ with $\sigma_i > 0$ for all $i \in \{1, 2, \ldots, d\}$ such that $u$ is analytically extendable to the complex polystrip $D_{\boldsymbol{\sigma}} = \{\mathbf{z} \in \mathbb{C}^d/2\pi\mathbb{Z}^d : |\operatorname{Im} z_i| < \sigma_i, i \in \{1, 2, \ldots, d\}\}$ and $u$ is bounded in $L^2$ on the boundary hyperplanes of $D_{\boldsymbol{\sigma}}$ so the Hardy norm*

$$\|u\|^2_{H\boldsymbol{\sigma}} := \sup_{\substack{|y_i| < \sigma_i \\ 1 \le i \le d}} \int_{\mathbb{T}^d} |u(\mathbf{x} + i\mathbf{y})|^2 \, d\mathbf{x} < \infty$$

*then there exists a LoRFS representation $g_\theta$ using terms up to frequency $K$ which is $\varepsilon$-close to $u$ in the sense that $\|u - g_\theta\|_{L^2} \le \varepsilon$ as long as $K = O(\log \frac{1}{\varepsilon})$.*

*Proof.* See the appendix. $\square$

**Discussion and Implications** The theoretical results established above provide rigorous justification for the efficiency of LoRFS in high-dimensional settings. While standard spectral methods on tensor product grids suffer from the curse of dimensionality and require $O(K^d)$ degrees of freedom, our analysis reveals that LoRFS effectively decouples the approximation error convergence from the dimension $d$. For functions in the three function spaces ($H^m$, $H^m_{\mathrm{mix}}$, analytic) discussed above, the complexity of

LoRFS is dominated by the rank $s$, which scales algebraically as $O(\varepsilon^{-2})$ regardless of $d$.

The theoretical guarantees established in this section reflect the intrinsic properties of many physical systems rather than serving as mere mathematical conveniences. For instance, elliptic and parabolic PDEs with analytic coefficients, fundamental to equilibrium problems and diffusion processes, often admit real-analytic solutions in the interior of the domain (Treves et al., 2022). In these settings, LoRFS leverages the underlying analyticity to achieve high precision with extremely compact models, effectively compressing the solution representation.

LoRFS's advantage extends to high-dimensional quantum mechanics. While electronic wavefunctions in high-dimensional quantum mechanics lack global analyticity due to particle interaction cusps, they are known to exhibit strong mixed smoothness properties (Yserentant, 2014). Our results suggest that LoRFS is particularly well-suited for such problems, as it captures high-dimensional correlations (entanglement) via the rank $s$ without the exponential explosion in parameters common to grid-based methods. This approach mirrors the efficiency of modern ansatz-based approaches in quantum chemistry.

Furthermore, the comparison between isotropic ($H^m$) and mixed ($H^m_{\text{mix}}$) smoothness highlights the versatility of LoRFS. While isotropic spaces capture general smoothness, mixed smoothness is the natural setting for sparse tensor approximations, analogous to the theory of Sparse Grids (Bungartz & Griebel, 2004). Our theorems confirm that LoRFS effectively exploits this structure to maximize parameter efficiency.

Finally, unlike discrete low-rank tensor decompositions (e.g., Tensor Train or CP-decomposition on fixed grids), LoRFS provides a *continuous* functional representation. This allows for exact, mesh-free differentiation and integration (as detailed in Section 4.1), eliminating discretization errors and enabling the solution of PDEs via variational formulations without the requirement of a background mesh.

### 4.3. Handling Non-Spectral Terms and Non-Periodic Boundaries

The analytic operations derived in Section 4.1 leverage the closure properties of the Fourier basis, implicitly assuming that the PDE terms are compatible with this spectral representation. To handle non-trigonometric components (e.g., general potentials or source terms) without losing analytic tractability, we employ a projection strategy. We approximate these non-conforming terms using auxiliary LoRFS models trained via standard regression and Monte Carlo sampling. Once trained, these approximations are substituted into the governing equation, converting the full PDE

into a form amenable to exact analytic evaluation.

While this approach reintroduces Monte Carlo sampling, it remains computationally efficient for two reasons. First, the constitutive terms in high-dimensional physics problems often reside in significantly lower-dimensional subspaces than the full solution (e.g., pairwise potentials in a many-body system). Second, the regression task—fitting a known target function—is significantly more stable and easier to optimize than minimizing a complex differential residual, which is known to suffer from pathological loss landscapes (Krishnapriyan et al., 2021; Rathore et al., 2024).

Finally, we address the constraint of periodic boundary conditions inherent to Fourier series. Non-periodic solutions can be represented by decoupling the basis frequency from the physical domain size $L$. By selecting a frequency $\omega < 2\pi/L$, we effectively embed the problem into a larger virtual torus $L' > L$, allowing the function values at the physical boundaries to differ. Alternatively, we can enforce specific boundary behaviors by multiplying the expansion by a smooth window function (itself approximated by LoRFS), effectively zeroing out contributions outside the region of interest while preserving analytic integrability.

As demonstrated in Section 5, this framework allows LoRFS to effectively solve PDEs with non-spectral terms and non-periodic boundaries, achieving state-of-the-art results. Further implementation details regarding these handling strategies are provided in the Appendix B.4.

## 5. Experiments

### 5.1. Experimental Setup

We conduct our evaluation on three standard PDE benchmarks: the convection, reaction, and wave equations. These systems are widely recognized in the literature for exhibiting specific failure modes which challenge standard coordinate-based networks (Krishnapriyan et al., 2021; Wu et al., 2024). To demonstrate that LoRFS is robust against these failure modes and capable of solving such equations even in complex regimes, we further evaluate it on extensions of these PDEs to higher spatial dimensions ($d > 2$). Detailed formulations of these PDEs and their high-dimensional extensions are provided in Appendix B.

We compare LoRFS against a comprehensive suite of state-of-the-art baselines, including vanilla PINN (Raissi et al., 2019), QRes (Bu & Karpatne, 2021), PINNsFormer (Zhao et al., 2023), KAN (Liu et al., 2024), PirateNet (Wang et al., 2024a), RoPINN (Wu et al., 2024), PINNMamba (Xu et al., 2025a), and PINN FP64 (Xu et al., 2025b).

In addition, we investigate the robustness of LoRFS in PDE settings involving high-order derivative operations, which often cause ill-conditioning and numerical instability. For

*Table 1.* **Performance comparison on standard PDE benchmarks known for exhibiting failure modes.** Baseline results are taken from previously reported works (Xu et al., 2025b). We report the relative $L^1$ (rMAE) and relative $L^2$ (rRMSE) errors (mean ± standard deviation) across the convection, reaction, and wave equations. LoRFS consistently achieves the lowest error, outperforming the previous state-of-the-art methods.

| | Convection | | Reaction | | Wave | |
|---|---|---|---|---|---|---|
| Model | rMAE | rRMSE | rMAE | rRMSE | rMAE | rRMSE |
| PINN | 0.6904 ± 0.0826 | 0.7640 ± 0.0694 | 0.9788 ± 0.0019 | 0.9778 ± 0.0018 | 0.2746 ± 0.0574 | 0.2837 ± 0.0571 |
| QRes | 0.7498 ± 0.0464 | 0.8184 ± 0.0382 | 0.9826 ± 0.0023 | 0.9830 ± 0.0026 | 0.5335 ± 0.1230 | 0.5273 ± 0.1172 |
| PINNsFormer | 0.0327 ± 0.0068 | 0.0435 ± 0.0073 | 0.0147 ± 0.0013 | 0.0296 ± 0.0027 | 0.3492 ± 0.0871 | 0.3571 ± 0.0872 |
| KAN | 0.6213 ± 0.0675 | 0.6985 ± 0.0701 | 0.0167 ± 0.0014 | 0.0312 ± 0.0034 | 0.1475 ± 0.0354 | 0.1489 ± 0.0357 |
| PirateNet | 0.9704 ± 0.1826 | 0.9740 ± 0.1894 | 0.0178 ± 0.0023 | 0.0443 ± 0.0064 | 0.2544 ± 0.0471 | 0.2637 ± 0.0480 |
| RoPINN | 0.6251 ± 0.0940 | 0.7204 ± 0.0941 | 0.0589 ± 0.0161 | 0.0965 ± 0.0310 | 0.0631 ± 0.0226 | 0.0642 ± 0.0238 |
| PINNMamba | 0.0184 ± 0.0037 | 0.0197 ± 0.0038 | 0.0092 ± 0.0017 | 0.0213 ± 0.0036 | 0.0193 ± 0.0033 | 0.0195 ± 0.0033 |
| PINN FP64 | 0.0059 ± 0.0013 | 0.0072 ± 0.0017 | 0.0271 ± 0.0063 | 0.0502 ± 0.0111 | 0.0080 ± 0.0032 | 0.0081 ± 0.0031 |
| **LoRFS** | **8.2E-06 ± 2.7E-06** | **9E-06 ± 2.9E-06** | **0.0002 ± 6.4E-06** | **0.0006 ± 1.4E-05** | **0.0033 ± 0.0017** | **0.0035 ± 0.0016** |

these experiments, we compare LoRFS to specialized methods designed to mitigate high-order differentiation issues by avoiding the exponential computation cost associated with building high-order computation graphs: Forward Laplacian (Li et al., 2023), STDE (Shi et al., 2024), and HTE (Hu et al., 2024).

To ensure a fair comparison of computational efficiency, we train LoRFS and all baseline models under a fixed computational budget for all experiments except Table 1. Specifically, each model is trained for exactly 1 hour using the L-BFGS optimizer on a single NVIDIA L4 GPU; Table 1 instead reports baseline results as reported in the original papers (literature numbers), while LoRFS results are from our implementation.

The primary evaluation metrics are the relative $L^2$ error (rRMSE) and the relative $L^1$ error (rMAE), defined as $\mathcal{E}_{L^2} = \frac{\|\hat{u}-u\|_2}{\|u\|_2}$ and $\mathcal{E}_{L^1} = \frac{\|\hat{u}-u\|_1}{\|u\|_1}$, respectively, where $\hat{u}$ denotes the model prediction and $u$ is the reference solution. These metrics are standard in the scientific machine learning literature (Xu et al., 2025a; Cho et al., 2024; Nam et al., 2024; Wu et al., 2024). Further implementation details, including baseline hyperparameters, and additional results are provided in the Appendix.

### 5.2. Main Results

Table 1 demonstrates that LoRFS successfully mitigates established optimization pathologies and significantly outperforms all baseline methods across the convection, reaction, and wave equations.

For the convection equation, LoRFS achieves a relative error lower by more than two orders of magnitude compared to the previous state-of-the-art method, PINN FP64. This substantial margin highlights the capability of LoRFS to accurately resolve PDEs with high-frequency oscillatory solutions, a regime where standard PINN based methods typically fail.

In the case of the reaction equation, LoRFS surpasses the best-performing baseline, PINNMamba, by over one order of magnitude. While PINNMamba addresses the "continuous-discrete mismatch" by introducing complex state-space models to recover continuous dynamics from discrete sub-sequences (Xu et al., 2025a), LoRFS fundamentally bypasses this limitation by eliminating stochastic collocation sampling entirely. By operating without the need to reconstruct continuity from discrete points, LoRFS achieves superior precision with greater efficiency.

Furthermore, the reaction equation serves as a critical robustness test, as it is governed by non-linear terms and possesses a solution that does not strictly align with the LoRFS spectral basis. The method's superior performance in this setting confirms that LoRFS is not merely overfitting to spectral-friendly problems, but generalizes robustly to non-spectral regimes.

A key advantage of LoRFS is its ability to scale efficiently to high-dimensional settings, whereas standard numerical methods suffer from the curse of dimensionality. We demonstrate this capability by evaluating LoRFS on high-dimensional extensions of the convection, wave, and reaction equations. Table 2 presents a comparison against the two strongest baselines, PINNMamba and PINN FP64.

*Table 2.* **Performance comparison on high-dimensional PDE extensions.** Each entry displays the relative $L^2$ error (rRMSE, top) and relative $L^1$ error (rMAE, bottom). LoRFS maintains high accuracy and outperforms the other baselines which fail to converge in this settings.

| High-Dim PDE | PINN_FP64 | PINNMamba | **LoRFS** |
|---|---|---|---|
| Convection | 0.6842 ± 0.0215 | 1.0095 ± 0.0562 | **0.0062 ± 0.0063** |
| | 0.6061 ± 0.0194 | 0.9857 ± 0.0628 | **0.0061 ± 0.0062** |
| Reaction | 1.0001 ± 0.0004 | 0.9996 ± 0.0012 | **0.0082 ± 0.005** |
| | 1.7445 ± 0.3049 | 1.7416 ± 0.5492 | **0.0409 ± 0.0408** |
| Wave | 0.8584 ± 0.0163 | 0.9978 ± 0.0006 | **5.9E-05 ± 3.6E-05** |
| | 0.8721 ± 0.0142 | 1.0525 ± 0.0044 | **7.1E-05 ± 4.3E-05** |

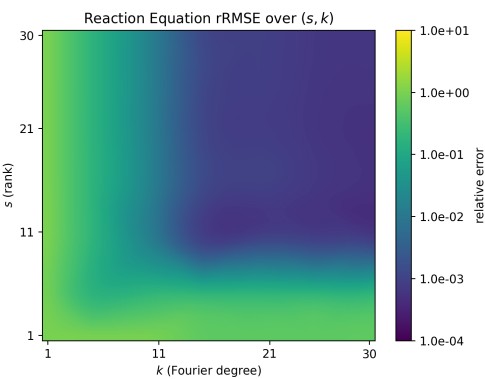

*Figure 2.* **Hyperparameter sensitivity analysis on the reaction equation.** The heatmap displays the relative $L^2$ error (rRMSE) across varying rank $s$ and frequency degree $K$. The results indicate a distinct capacity threshold, beyond which increasing model complexity yields diminishing returns in accuracy.

The results indicate a stark contrast in performance: while the baseline methods fail to converge in these high-dimensional settings, LoRFS maintains high accuracy.

Notably, PINNMamba yields the highest errors in this regime; this is attributed to its significant computational overhead, which prevents adequate convergence within the strict one-hour computational budget compared to the more lightweight PINN FP64.

### 5.3. Robustness and Stability Analysis

We first investigate the robustness of LoRFS in PDE settings characterized by high-order derivative operations, which are prone to inducing ill-conditioning and numerical instability. We demonstrate that LoRFS effectively mitigates these challenges by leveraging analytical differentiation via its spectral representation, thereby avoiding the compounding approximation errors inherent to finite differences or standard automatic differentiation in high-order settings.

Table 3 presents the results for the biharmonic equation, which involves a fourth-order operator. LoRFS achieves the highest accuracy among all compared methods. This performance is consistent with the method's design, which computes the derivative operator exactly rather than through approximation. Notably, the second-best performing method is the Forward Laplacian, which also utilizes a deterministic approach. In contrast, methods such as HTE and STDE rely on stochastic Monte Carlo estimators to approximate the biharmonic operator. Because these baselines depend on sampling points within the domain, they introduce significant variance and yield higher errors. These findings underscore the capability of LoRFS to robustly solve high-order differential equations without the instability introduced by stochastic gradient estimation.

*Table 3.* **Robustness on high-order PDEs.** On the 4th-order biharmonic equation, LoRFS outperforms both stochastic and deterministic baselines, demonstrating the stability of its analytical differentiation.

| Method | rMAE | rRMSE |
|---|---|---|
| HTE | 1.97E-02 $\pm$ 7.92E-4 | 2.02E-02 $\pm$ 8.71E-4 |
| STDE | 2.29E-02 $\pm$ 1.8E-03 | 2.36E-2 $\pm$ 1.65E-03 |
| Forward Laplacian | 7.8E-03 $\pm$ 5.9E-04 | 8E-03 $\pm$ 6.3E-04 |
| **LoRFS** | **4.6E-04 $\pm$ 2E-04** | **5.15E-04 $\pm$ 2.4E-04** |

Complementing the high-order analysis, we evaluate the stability of LoRFS with respect to its key hyperparameters: the rank $s$ and the maximum frequency $K$. We conduct this sensitivity analysis on the reaction equation, a non-linear problem chosen because its solution structure does not trivially align with the LoRFS spectral basis. Figure 2 illustrates the performance (relative RMSE) across various configurations of $s$ and $K$. The results reveal a clear capacity threshold where the model requires at least $K > 5$ and $s > 5$ for adequate approximation. Beyond these values, we observe diminishing returns; this plateau confirms the method's stability, demonstrating that it remains robust even when the target function lacks ideal alignment with the spectral basis.

## 6. Limitations

LoRFS is designed for regimes in which the solution and the relevant PDE terms admit a compact low-rank Fourier representation. This includes many smooth, oscillatory, and high-dimensional problems, but it does not make LoRFS a universal replacement for all PINN-style solvers. When the target solution is not well compressed by a modest number of separable Fourier factors, the required rank $s$ and possibly the frequency cutoff $K$ may grow substantially, reducing the computational advantage over more general representations. This issue is expected to become more pronounced for long-time dynamics, chaotic systems, and turbulent flows where energy may remain distributed across many spatial and temporal frequencies. To probe this setting beyond the main benchmark suite, we evaluated LoRFS on a Navier-Stokes problem and obtained encouraging results (appendix B.5), but this does not establish efficiency in fully turbulent or strongly chaotic regimes.

## 7. Conclusion

By parameterizing solutions as separable, low-rank Fourier series, LoRFS leverages a global spectral basis that naturally resolves high-frequency oscillatory patterns, thereby overcoming the spectral bias inherent to standard neural networks. Moreover, this representation replaces the noisy, sample-inefficient collocation paradigm with exact, closed-form evaluation of loss integrals and derivatives. This ar-

chitectural shift effectively eliminates quadrature error and gradient variance, which are primary drivers of optimization failure in stiff and oscillatory regimes.

Our theoretical analysis establishes that LoRFS attains favorable high-dimensional scaling: its convergence rates depend primarily on the rank (and the smoothness of the target) rather than growing exponentially with the spatial dimension. Empirically, these advantages translate into state-of-the-art performance: LoRFS reduces errors by orders of magnitude on standard failure-mode benchmarks (convection, reaction, and wave equations) compared to leading baselines like PINNMamba and PINN FP64. Crucially, we demonstrated that LoRFS maintains robust performance in high-dimensional settings where PINN-based methods fail to converge. Furthermore, LoRFS exhibits exceptional stability in solving high-order PDEs, such as the biharmonic equation, by leveraging deterministic analytical differentiation.

## Impact Statement

This paper presents work whose goal is to advance the field of Machine Learning. There are many potential societal consequences of our work, none which we feel must be specifically highlighted here.

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

## A. Theoretical Section Proofs

Before stating any of the relevant theorems, it is necessary to introduce some notation. We begin by considering functions $\breve{u}$ which can be expressed as expansions in terms of basis functions:

$$\breve{u} = \sum_{\nu \in \mathcal{F}} c_\nu \phi_\nu \tag{12}$$

where $\Omega$ is the hypertorus $\mathbb{T}^d$, $\phi_\nu : \Omega \to \mathbb{C}$ are basis functions, and $c_\nu \in \mathbb{C}$ are the corresponding coefficients (with the appropriate constraints to ensure the range of $\breve{u}$ is real), with $\mathcal{F}$ being a countable index set. In the following, any subset of $\mathcal{F}$, either explicit or implied (e.g., domains of summation) is assumed to be constrained such that the range of $\breve{u}$ is real. Constraining the range of $\breve{u}$ to the reals is solely for the sake of simplicity and to parallel the software implementation; all of the proofs can be extended to complex-valued functions.

We will be interested in understanding how accurately a given function $u$ can be expressed as in Equation (12). The following definitions will prove useful in describing the quality of the basis expansion approximation.

**Definition A.1.** Let $(\Lambda_n)_{n \geq 1}$ be a sequence of finite subsets of the index set $\mathcal{F}$. This sequence is called an exhaustion of $\mathcal{F}$ if, for every $\nu \in \mathcal{F}$, there exists an integer $n_0$ such that $\nu \in \Lambda_n$ for all $n \geq n_0$.

**Definition A.2.** The series in Equation (12) is said to conditionally converge to $u$ under a given norm $\| \cdot \|$ if there exists an exhaustion $(\Lambda_n)_{n \geq 1}$ of $\mathcal{F}$ such that:

$$\lim_{n \to \infty} \left\| u - \sum_{\nu \in \Lambda_n} c_\nu \phi_\nu \right\| = 0. \tag{13}$$

**Definition A.3.** The series in Equation (12) is said to unconditionally converge to $u$ in the same norm if and only if equation (13) is satisfied for every exhaustion $(\Lambda_n)_{n \geq 1}$ of $\mathcal{F}$.

Finally, we have a standard definition of $\ell^p$, which we state here for the reader's convenience:

**Definition A.4.** For $0 < p < \infty$, the space $\ell^p$ is defined as the subset of $\mathbb{C}^{\mathbb{N}}$, the space of all sequences, consisting of elements $(x_n)_{n \in \mathbb{N}}$ that satisfy the condition:

$$\sum_{n=1}^{\infty} |x_n|^p < \infty. \tag{14}$$

Armed with the previous definitions, we are ready to attack the problem of the convergence rate of approximations based on the LoRFS model class. To prove our main result, we will need three results. The first theorem is proven in (Cohen & DeVore, 2015), while we prove the following two lemmata.

Let us begin with the first result we will need. Theorem 3.3 from (Cohen & DeVore, 2015) states the following:

**Theorem A.5.** *Let $(c_\nu)_{\nu \in \mathcal{F}}$ be an orthonormal basis of $L^2(U, \mu)$ for some given measure $\mu$ on $U$, and let $u \in L^2(U, V, \mu)$. Then, the inner products*

$$u_\nu := \int u(y) \phi_\nu(y) d\mu(y), \ \nu \in \mathcal{F} \tag{15}$$

*are elements of $V$, and the series in Equation (12) converges unconditionally towards $u$ in $L^2(U, V, \mu)$, with the error given by*

$$\left\| u - \sum_{\nu \in \Lambda_n} c_\nu \phi_\nu \right\|_{L^2(U,V,\mu)} \leq \left( \sum_{\nu \notin \Lambda_n} |c_\nu|_V^2 \right)^{\frac{1}{2}}. \tag{16}$$

*for any exhaustion $(\Lambda_n)_{n \geq 1}$.*

In our case of a Fourier basis, we will of course apply Theorem (A.5) with the uniform probability measure over $\mathbb{T}^d$.

The second result we require is a generalization of Stechkin's Lemma; indeed, Stechkin's Lemma corresponds to the special case $q = 2$. We provide the proof in this instance.

**Lemma A.6.** *Let $0 < p < q < \infty$ and let $(c_\nu)_{\nu \in \mathcal{F}} \in \ell^p(\mathcal{F})$ be a sequence of complex numbers. Then, if $\Lambda_n$ is a set of indices which corresponds to the $n$ largest values of $|c_\nu|$, one has*

$$\left( \sum_{\nu \notin \Lambda_n} |c_\nu|^q \right)^{\frac{1}{q}} \leq (n+1)^{-\left(\frac{1}{p} - \frac{1}{q}\right)} \left( \sum_{\nu \in \mathcal{F}} |c_\nu|^p \right)^{\frac{1}{p}} \tag{17}$$

*Proof.* Let $(c_k)_{k \geq 1}$ be the decreasing rearrangement of the sequence $(c_\nu)_{\nu \in \mathcal{F}}$, sorted according to descending modulus. From the definition of $\Lambda_n$, we have:

$$\sum_{\nu \notin \Lambda_n} |c_\nu|^q = \sum_{k \geq n+1} |c_k|^q \leq |c_{n+1}|^{q-p} \sum_{k \geq n+1} |c_k|^p \leq |c_{n+1}|^{q-p} \sum_{\nu \in \mathcal{F}} |c_\nu|^p \tag{18}$$

On the other hand, we also have:

$$(n+1)|c_{n+1}|^p \leq \sum_{k=0}^{n+1} |c_k|^p \leq \sum_{\nu \in \mathcal{F}} |c_\nu|^p \tag{19}$$

Now, dividing this inequality by $(n+1)$, we obtain:

$$|c_{n+1}|^p \leq \frac{\sum_{\nu \in \mathcal{F}} |c_\nu|^p}{n+1}. \tag{20}$$

Substituting this bound into the previous inequality, we have:

$$\sum_{\nu \notin \Lambda_n} |c_\nu|^q \leq (|c_{n+1}|^p)^{\frac{q-p}{p}} \sum_{\nu \in \mathcal{F}} |c_\nu|^p \leq \left( \frac{\sum_{\nu \in \mathcal{F}} |c_\nu|^p}{n+1} \right)^{\frac{q-p}{p}} \sum_{\nu \in \mathcal{F}} |c_\nu|^p \tag{21}$$

Finally, raising both sides to the power of $1/q$, we recover the desired inequality. $\qquad \square$

To obtain concrete results concerning the summability of $c_\nu$ we take the $\{\phi_\nu\}$ to be the Fourier basis, i.e., tensor products of complex exponentials $e^{inx}$ and we will need to define $\mathcal{F} := \mathbb{Z}^d$ and define various relations and operations on its elements.

**Definition A.7.** Let $\boldsymbol{\alpha}$ and $\boldsymbol{\beta}$ be elements of $\mathcal{F} := \mathbb{Z}^d$, and $\gamma$ be a scalar element from $\mathbb{Z}$.

(1) $|\boldsymbol{\alpha}| := (|\alpha_1|, |\alpha_2|, \ldots, |\alpha_d|)$.

(2) $\boldsymbol{\alpha} \odot \boldsymbol{\beta} := (\alpha_1\beta_1, \alpha_2\beta_2, \ldots, \alpha_d\beta_d)$.

(3) $\boldsymbol{\alpha} \cdot \boldsymbol{\beta} := \sum_{i=1}^{d} \alpha_i \beta_i$.

(4) $\boldsymbol{\alpha} \leq \gamma := \alpha_i \leq \gamma$ for all $i = 1, 2, \ldots, d$.

(5) $\boldsymbol{\alpha} < \gamma := \boldsymbol{\alpha} \leq \gamma$ and in addition $\exists i$ s.t. $\alpha_i < \gamma$.

(6) $\boldsymbol{\alpha} \not\leq \gamma := \exists i$ s.t. $\alpha_i > \gamma$.

*Proof of Theorem 4.5.* Equation (5.8.4) in (Canuto et al., 2006) applies, so

$$\|u - P_K u\|_{H^l(\Omega)} \leq C N^{l-m} |u|_{H^m(\Omega)} \quad \text{for } 0 \leq l \leq m$$

and the result then follows after setting $l = 0$ and inverting the relation between $K$ and $\varepsilon$. $\qquad \square$

*Proof of Theorem 4.6.* From the theorem from Section 2.7.1 from (Schmeisser & Triebel, 1987) while taking $p = q = 2$,

$$\varepsilon = \|u - \sum_{|\nu_i| \leq K} c_\nu \phi_\nu\| = O(K^{-m}).$$

The result then follows from inverting the relation between $K$ and $\varepsilon$. $\qquad \square$

*Proof of Theorem 4.7.* Let $S_d = \{-1, +1\}^d$ be the set of all $d$-dimensional binary vectors. The corners of the polystrip are then $V_d = \{\mathbf{s} \odot \boldsymbol{\sigma} = (s_1\sigma_1, s_2\sigma_2, \ldots, s_d\sigma_d) : \mathbf{s} \in S_d\}$. Let

$$\mathrm{sgn}^*(x) = \begin{cases} +1 & \text{if } x \geq 0 \\ -1 & \text{if } x < 0 \end{cases}$$

so that for every $\mathbf{k} \in \mathbb{Z}^d$, the vector $\mathrm{NegSgn}(\mathbf{k}) := (-\mathrm{sgn}^*(k_1), -\mathrm{sgn}^*(k_2), \ldots, -\mathrm{sgn}^*(k_d))$ is a member of $S_d$. Because the Hardy norm of $u$ is well-defined,

$$\sum_{\mathbf{s} \in S_d} \int_{\mathbb{T}^d} |u(\mathbf{x} + i(\mathbf{s} \odot \boldsymbol{\sigma})))|^2 \, d\mathbf{x} \leq 2^d \|u\|_{H_{\boldsymbol{\sigma}}^2} < \infty$$

so by using the Fourier transform of $u$ in the polystrip

$$u(\mathbf{x} + i\mathbf{y}) = \sum_{\mathbf{k} \in \mathbb{Z}^d} c_{\mathbf{k}} e^{i\mathbf{k} \cdot (\mathbf{x} + i\mathbf{y})} = \sum_{\mathbf{k} \in \mathbb{Z}^d} c_{\mathbf{k}} e^{-\mathbf{k} \cdot \mathbf{y}} e^{i\mathbf{k} \cdot \mathbf{x}}$$

and Parseval's theorem, we have

$$\sum_{\mathbf{s} \in S_d} \sum_{\mathbf{k} \in \mathbb{Z}^d} |c_{\mathbf{k}}|^2 e^{-2\mathbf{k} \cdot (\mathbf{s} \odot \boldsymbol{\sigma})} < \infty$$

and

$$\sum_{\mathbf{k} \in \mathbb{Z}^d} |c_{\mathbf{k}}|^2 e^{-2\mathbf{k} \cdot (\mathrm{NegSgn}(\mathbf{k}) \odot \boldsymbol{\sigma})} \leq \sum_{\mathbf{s} \in S_d} \sum_{\mathbf{k} \in \mathbb{Z}^d} |c_{\mathbf{k}}|^2 e^{-2\mathbf{k} \cdot (\mathbf{s} \odot \boldsymbol{\sigma})}$$

by changing the order of summation and discarding all the elements of $S_d$ other than $\mathrm{NegSgn}(\mathbf{k})$ in the sum over $S_d$. Since

$$\sum_{\mathbf{k} \in \mathbb{Z}^d} |c_{\mathbf{k}}|^2 e^{-2\mathbf{k} \cdot (\mathrm{NegSgn}(\mathbf{k}) \odot \boldsymbol{\sigma})} = \sum_{\mathbf{k} \in \mathbb{Z}^d} |c_{\mathbf{k}}|^2 e^{2|\mathbf{k}| \cdot \boldsymbol{\sigma}} = \|u\|_{G_{\boldsymbol{\sigma},0}}^2 \tag{22}$$

it is clear that $u$ is a member of the Gevrey space $G_{\boldsymbol{\sigma},0}$. Let $P_K u$ be the projection of the function $u$ onto the hypercube frequency set $\{\mathbf{k} \in \mathbb{Z}^d : |\mathbf{k}| \leq K\}$. Then the truncation error satisfies

$$\varepsilon^2 = \|u - P_K u\|_{L^2}^2 = \sum_{\mathbf{k} \not\leq K} |c_{\mathbf{k}}|^2 \leq \sum_{i=1}^{d} \sum_{\substack{\mathbf{k} \in \mathbb{Z}^d \\ k_i > K}} |c_{\mathbf{k}}|^2.$$

Considering only one element over the dimensions,

$$\sum_{\substack{\mathbf{k} \in \mathbb{Z}^d \\ k_i > K}} |c_{\mathbf{k}}|^2 = \sum_{\substack{\mathbf{k} \in \mathbb{Z}^d \\ k_i > K}} |c_{\mathbf{k}}|^2 \frac{e^{2\boldsymbol{\sigma} \cdot |\mathbf{k}|}}{e^{2\boldsymbol{\sigma} \cdot |\mathbf{k}|}} \leq e^{-2\sigma_i K} \sum_{\substack{\mathbf{k} \in \mathbb{Z}^d \\ k_i > K}} |c_{\mathbf{k}}|^2 e^{2\boldsymbol{\sigma} \cdot |\mathbf{k}|} \leq e^{-2\sigma_i K} \sum_{\mathbf{k} \in \mathbb{Z}^d} |c_{\mathbf{k}}|^2 e^{2\boldsymbol{\sigma} \cdot |\mathbf{k}|} = e^{-2\sigma_i K} \|u\|_{G_{\boldsymbol{\sigma},0}}^2$$

so

$$\sum_{\mathbf{k} \not\leq K} |c_{\mathbf{k}}|^2 \leq \|u\|_{G_{\boldsymbol{\sigma},0}}^2 \sum_{i=1}^{d} e^{-2\sigma_i K}$$

$$\varepsilon^2 = \|u - P_K u\|_{L^2}^2 \leq \|u\|_{G_{\boldsymbol{\sigma},0}}^2 \sum_{i=1}^{d} e^{-2\sigma_i K} \leq \|u\|_{G_{\boldsymbol{\sigma},0}}^2 d e^{-2(\min_i \sigma_i) K}$$

The result then follows from taking square roots and inverting the relation between $K$ and $\varepsilon$. $\qquad \square$

**Theorem A.8.** *If there exists a vector $\boldsymbol{\sigma} = (\sigma_1, \sigma_2, \ldots, \sigma_d)$ with $\sigma_i > 0$ for all $i \in \{1, 2, \ldots, d\}$ such that $f$ is analytically extendable to the complex polystrip $D_{\boldsymbol{\sigma}} = \{\mathbf{z} \in \mathbb{C}^d / 2\pi \mathbb{Z}^d : |\mathrm{Im}\, z_i| < \sigma_i, i \in \{1, 2, \ldots, d\}\}$ and $f$ is bounded in $L^2$ on the boundary hyperplanes of $D_{\boldsymbol{\sigma}}$ so the Hardy norm*

$$\|f\|_{H_{\boldsymbol{\sigma}}}^2 := \sup_{\substack{|y_i| < \sigma_i \\ 1 \leq i \leq d}} \int_{\mathbb{T}^d} |f(\mathbf{x} + i\mathbf{y})|^2 \, d\mathbf{x} < \infty$$

*is well-defined, and $\{c_\nu\}$ are the Fourier coefficients of $f$, then*

$$\sum_{\nu \in \mathcal{F}} |c_\nu| < \infty. \tag{23}$$

*Proof.* As previously shown in the proof of Theorem 4.7, equation 22, $f$ is in the Gevrey space $G_{\boldsymbol{\sigma},0}$. Since

$$\sum_{\mathbf{k}^* \in \mathbb{Z}^d} |c_{\mathbf{k}^*}|^2 e^{2|\mathbf{k}^*| \cdot \boldsymbol{\sigma}} = \|u\|_{G_{\boldsymbol{\sigma},0}}^2$$

we have the following bound for $c_{\mathbf{k}}$

$$|c_{\mathbf{k}}|^2 e^{2\boldsymbol{\sigma} \cdot |\mathbf{k}|} \leq \sum_{\mathbf{k}^* \in \mathbb{Z}^d} |c_{\mathbf{k}^*}|^2 e^{2|\mathbf{k}^*| \cdot \boldsymbol{\sigma}} = \|u\|_{G_{\boldsymbol{\sigma},0}}^2$$

$$|c_{\mathbf{k}}| e^{\boldsymbol{\sigma} \cdot |\mathbf{k}|} \leq \|u\|_{G_{\boldsymbol{\sigma},0}} \implies |c_{\mathbf{k}}| \leq \|u\|_{G_{\boldsymbol{\sigma},0}} e^{-\boldsymbol{\sigma} \cdot |\mathbf{k}|}$$

Therefore,

$$\sum_{\mathbf{k} \in \mathbb{Z}^d} |c_{\mathbf{k}}| \leq \sum_{\mathbf{k} \in \mathbb{Z}^d} \|u\|_{G_{\boldsymbol{\sigma},0}} e^{-\boldsymbol{\sigma} \cdot |\mathbf{k}|} = \|u\|_{G_{\boldsymbol{\sigma},0}} \sum_{\mathbf{k} \in \mathbb{Z}^d} \prod_{i=1}^{d} e^{-\sigma_i |k_i|}$$

$$\|u\|_{G_{\boldsymbol{\sigma},0}} \sum_{\mathbf{k} \in \mathbb{Z}^d} \prod_{i=1}^{d} e^{-\sigma_i |k_i|} = \|u\|_{G_{\boldsymbol{\sigma},0}} \prod_{i=1}^{d} \sum_{n=-\infty}^{\infty} e^{-\sigma_i |n|} = \|u\|_{G_{\boldsymbol{\sigma},0}} \prod_{i=1}^{d} \frac{1 + e^{-\sigma_i}}{1 - e^{-\sigma_i}} < \infty$$

and we are finished. $\square$

**Theorem A.9.** *If $f$ is in the Sobolev space $H^m(\Omega)$ with $m > d/2$ and $\{c_\nu\}$ are the Fourier coefficients of $f$, then*

$$\sum_{\nu \in \mathcal{F}} |c_\nu| < \infty. \tag{24}$$

*Proof.*

$$\sum_{\nu \in \mathcal{F}} |c_\nu| = \sum_{\nu \in \mathcal{F}} \left( |c_\nu|(1 + |\nu|^2)^{m/2} \frac{1}{(1 + |\nu|^2)^{m/2}} \right)$$

so by Cauchy-Schwarz

$$\sum_{\nu \in \mathcal{F}} |c_\nu| \leq \sqrt{\sum_{\nu \in \mathcal{F}} |c_\nu|^2 (1 + |\nu|^2)^m} \sqrt{\sum_{\nu \in \mathcal{F}} \frac{1}{(1 + |\nu|^2)^m}}$$

but the first square root term is just the Sobolev norm and the sum in the second term converges if $m > d/2$. $\square$

**Theorem A.10.** *If $f$ is in the mixed smoothness Sobolev space $H_{\text{mix}}^m(\Omega)$ with $m > 1/2$ and $\{c_\nu\}$ are the Fourier coefficients of $f$, then*

$$\sum_{\nu \in \mathcal{F}} |c_\nu| < \infty. \tag{25}$$

*Proof.* The equivalence between $H_{\text{mix}}^m(\Omega = \mathbb{T}^d)$ and the $d$-times repeated tensor product $H^m(\mathbb{T}) \otimes H^m(\mathbb{T}) \otimes \cdots \otimes H^m(\mathbb{T})$ is proven in (Schmeisser & Triebel, 1987) in Section 2.3.1. It is well-known that $H^m(\mathbb{T})$ is in the Wiener algebra $A(\mathbb{T})$ iff $m > 1/2$. Since tensor products of spaces in the Wiener algebra are in the Wiener algebra of the corresponding tensor domain, all elements of $H_{\text{mix}}^m(\Omega)$ with $m > 1/2$ are in the Wiener algebra over $\mathbb{T}^d$ and therefore satisfy the stated summability condition. $\square$

We are now ready to prove Theorems 4.2, 4.3 and 4.4.

*Proof of Theorems 4.2, 4.3 and 4.4.* Theorem A.5 can be applied to $f$ using the Fourier basis $\{\phi_\nu\}$ and combined with Lemma A.6 where $n = s$, $p = 1$ and $q = 2$ to obtain

$$\epsilon = \left\| u - \sum_{\nu \in \Lambda_n} c_\nu \phi_\nu \right\|_{L^2(U,V,\mu)} \leq (s+1)^{-\left(1-\frac{1}{2}\right)} \sum_{\nu \in \mathcal{F}} |c_\nu|$$

Application of the appropriate theorem from Theorems A.9, A.10 or A.8 then shows that there exists a constant $C$ which does not depend on $s$ such that

$$\epsilon \leq C(s+1)^{-\frac{1}{2}}$$

which can be rearranged to the desired result. ☐

## B. Experimental Details

### B.1. Training Details

All experiments are repeated over five random seeds, and we report the mean and standard deviation across runs. LoRFS, PINN (FP64), and PINNMamba are trained with L-BFGS using a learning rate of 1 and the strong Wolfe line search; all three are run in FP64. STDE, HTE, and Forward Laplacian are trained with Adam (Kingma, 2014) using a learning rate of $10^{-3}$, following the settings used in recent STDE work. Each method is allocated a fixed budget of 60 minutes per run on the same NVIDIA L4 GPU.

Whenever applicable, we reproduce the model architectures exactly as specified in the corresponding publications. Hyperparameter values are taken directly from the original works to ensure consistency across baselines (Table 4), and the number of collocation points used by each method is reported in the same table. By matching architectures, hyperparameters, hardware, and compute budgets, we aim to attribute observed performance differences to methodological choices rather than tuning or experimental design.

### B.2. Metrics

We evaluate solution accuracy by comparing the predicted solution $u_\theta$ to the ground truth $u_*$ using the relative Mean Absolute Error (rMAE) and relative Root Mean Squared Error (rRMSE). Both metrics are computed over a fixed set of evaluation points $\mathcal{S}$, with $|\mathcal{S}| = 10,000$ for all benchmarks:

$$\text{rMAE} = \frac{\sum_{x \in \mathcal{S}} |u_\theta(x) - u_*(x)|}{\sum_{x \in \mathcal{S}} |u_*(x)|}, \tag{26}$$

$$\text{rRMSE} = \sqrt{\frac{\sum_{x \in \mathcal{S}} (u_\theta(x) - u_*(x))^2}{\sum_{x \in \mathcal{S}} (u_*(x))^2}}. \tag{27}$$

These relative metrics are scale-invariant, enabling fair comparisons across PDEs with different solution magnitudes.

*Table 4.* Model hyperparameters and training setup.

| Model | Hyperparameters | Parameters | Collocation Points |
|---|---|---|---|
| PINN (FP64) | Hidden layers = 4; Hidden size = 128 | 33k | 20k |
| PINNMamba | # encoders = 1; Embedding size = 32; $\Delta, B, C$ width = 8; Hidden size = 512; Sequence length $k = 7$; Sequence interval $\Delta t = 10^{-2}$ | 285k | 2k |
| STDE | Hidden layers = 4; Hidden size = 128; Biharmonic estimation points = 100 | 33k | 200 |
| HTE | Hidden layers = 4; Hidden size = 128; Biharmonic estimation points = 100 | 33k | 200 |
| Forward Laplacian | Hidden layers = 4; Hidden size = 128 | 33k | 200 |
| LoRFS | $K = 30$; $s = 30$ | 9k | – |

## B.3. PDE Benchmarks

In this section, we provide the exact formulations of the Partial Differential Equations (PDEs) used in the experiments. For the high-dimensional experiments presented in Table 2, we set the spatial dimension to $d = 9$, otherwise, we use the standard setting with $d = 1$.

### B.3.1. CONVECTION EQUATION

We consider the high-dimensional linear convection (transport) equation. This benchmark tests the model's ability to resolve multiscale propagation features without numerical dissipation.

$$\frac{\partial u}{\partial t} + \beta \sum_{i=1}^{d} \frac{\partial u}{\partial x_i} = 0, \quad x \in [0, 2\pi]^d, \, t \in [0, 1] \tag{28}$$

**Parameters:** We use a convection coefficient $\beta = 50$ for the standard case and $\beta = 15$ for the high-dimensional case.
**Initial Condition:** The system is initialized with a sum of sines:

$$u(x, 0) = \sum_{i=1}^{d} \sin(x_i). \tag{29}$$

**Boundary Conditions:** We enforce periodic boundary conditions across all spatial dimensions $x_i \in [0, 2\pi]$.
**Solution:** The analytic solution is a traveling wave:

$$u(x, t) = \sum_{i=1}^{d} \sin(x_i - \beta t). \tag{30}$$

### B.3.2. REACTION EQUATION

We solve the logistic reaction equation coupled with a high-dimensional spatial distribution. This problem is stiff due to the exponential growth/decay dynamics controlled by the reaction rate $\rho$.

$$\frac{\partial u}{\partial t} - \rho u(1 - u) = 0, \quad x \in [0, 2\pi]^d, \, t \in [0, 1] \tag{31}$$

**Parameters:** We set the reaction rate to $\rho = 5$ for the standard case and $\rho = 1$ for the high-dimensional case.
**Initial Condition:** The spatial dependency is introduced via the initial condition, which is a Gaussian bump centered at $\pi$:

$$u(x, 0) = h(x) = \exp\left(-\frac{\|x - \pi\|^2}{2(\pi/4)^2}\right). \tag{32}$$

**Boundary Conditions:** We enforce periodic boundary conditions across all spatial dimensions $x_i \in [0, 2\pi]$.
**Solution:** The analytic solution is given by the sigmoid function:

$$u(x, t) = \frac{h(x)e^{\rho t}}{h(x)e^{\rho t} + 1 - h(x)}. \tag{33}$$

### B.3.3. WAVE EQUATION

We consider the high-dimensional wave equation. The wave speed is scaled by the dimension to maintain consistent wave propagation characteristics across dimensions.

$$\frac{\partial^2 u}{\partial t^2} - c^2 \Delta u = 0, \quad x \in [0, 1]^d, \, t \in [0, 1] \tag{34}$$

**Parameters:** We set the wave speed to $c = \sqrt{\frac{4}{d}}$, and use $\beta_2 = 3$ for the standard case and $\beta_2 = 1$ for the high-dimensional case.

**Initial Conditions:** We impose an initial displacement composed of a superposition of high-frequency standing waves, with zero initial velocity:

$$u(x,0) = \prod_{i=1}^{d} \sin(\pi x_i) + 0.5 \prod_{i=1}^{d} \sin(\beta_2 \pi x_i), \tag{35}$$

$$\frac{\partial u}{\partial t}(x,0) = 0. \tag{36}$$

**Boundary Conditions:** We enforce homogeneous Dirichlet boundary conditions, $u(x,t) = 0$, on the boundary $\partial \Omega$ of the unit hypercube.

**Solution:** The analytic solution is:

$$u(x,t) = \left( \prod_{i=1}^{d} \sin(\pi x_i) \right) \cos(2\pi t) + 0.5 \left( \prod_{i=1}^{d} \sin(\beta_2 \pi x_i) \right) \cos(2\beta_2 \pi t). \tag{37}$$

### B.3.4. BIHARMONIC EQUATION

We solve a fourth-order PDE, which typically poses significant challenges for automatic differentiation due to high computational graph cost.

$$\Delta^2 u + \omega^4 u = f(x), \quad x \in [0,1]^d \tag{38}$$

**Parameters:** We evaluate only the high-dimensional case for $d = 10$ and set the frequency to $\omega = 2\pi$.

**Source Term:** The source term is constructed to support a solution composed of a sum of cosines:

$$f(x) = 2\omega^4 \sum_{i=1}^{d} \cos(\omega x_i). \tag{39}$$

**Boundary Conditions:** We enforce periodic boundary conditions on the domain $[0,1]^d$.

**Solution:** The analytic solution is:

$$u(x) = \sum_{i=1}^{d} \cos(\omega x_i). \tag{40}$$

### B.4. Non-Spectral Terms and Non-Periodic Boundaries

**Non-spectral terms.** As discussed in Section 4.3, when the differential operator contains non-spectral terms (or when boundary/initial data are not exactly representable in the Fourier tensor basis), we handle these components via a projection step. Since this projection is approximate, it can become a dominant source of error and therefore must be quantified.

Among our benchmarks, the Reaction equation is the only case that requires projection: the initial condition at $t = 0$ is not exactly expressible as a finite sum of Fourier tensor products. To isolate the impact of this approximation, we perform an ablation in which we control the quality of the projected initial condition and measure the downstream effect on the final PDE solution. Concretely, we fit an *auxiliary* LoRFS model to the initial condition and use its output as the projected input to the *main* LoRFS PDE solver. Because the initial condition is separable, we fix the rank to $s = 1$ and sweep only the Fourier degree $K$, which directly controls the approximation capacity of the auxiliary projection model.

Table 5 reveals a strong coupling between *projection error* and *solution error*: coarse projections (large auxiliary rRMSE) yield large Reaction errors, while improving the projection rapidly decreases both rRMSE and rMAE of the PDE solution. Overall, increasing $K$ improves projection fidelity and correspondingly reduces Reaction error, with the largest gains up to $K \approx 10$. Beyond that point, performance largely saturates, suggesting that once the initial condition is represented accurately enough, the remaining error is dominated by other factors such as solver optimization difficulty rather than by projection itself.

Finally, Table 1 shows that projection is a key enabler for extending LoRFS beyond strictly spectral settings: once the projected terms are approximated with modest fidelity, LoRFS remains accurate. Overall, projection provides a simple, controllable way to handle non-spectral components without becoming the limiting factor.

*Table 5.* Effect of Fourier degree $K$ (rank fixed to $s = 1$) on (i) the auxiliary LoRFS approximation of the non-spectral initial condition and (ii) downstream accuracy on the Reaction benchmark.

| $(s, K)$ | Auxiliary projection rRMSE | Reaction rRMSE | Reaction rMAE |
|---|---|---|---|
| 1,1 | 0.7481 | 0.6428 | 0.6033 |
| 1,2 | 0.2818 | 0.2135 | 0.1776 |
| 1,3 | 0.0590 | 0.3017 | 0.1908 |
| 1,4 | 0.0068 | 0.0464 | 0.0245 |
| 1,5 | $4.60 \times 10^{-4}$ | 0.0055 | 0.0024 |
| 1,10 | $2.75 \times 10^{-5}$ | $9.00 \times 10^{-4}$ | $3.00 \times 10^{-4}$ |
| 1,20 | $1.08 \times 10^{-5}$ | $5.80 \times 10^{-4}$ | $1.60 \times 10^{-4}$ |
| 1,30 | $6.09 \times 10^{-6}$ | $5.70 \times 10^{-4}$ | $1.50 \times 10^{-4}$ |
| 1,40 | $3.74 \times 10^{-6}$ | $6.20 \times 10^{-4}$ | $1.70 \times 10^{-4}$ |
| 1,50 | $2.84 \times 10^{-6}$ | $6.00 \times 10^{-4}$ | $2.00 \times 10^{-4}$ |

**Non-periodic boundaries.** Fourier representations are periodic on their underlying period. To represent non-periodic solutions on a physical domain $\Omega = \prod_{j=1}^{d}[a_j, b_j]$ with side-lengths $L_j = b_j - a_j$, we can use two equivalent strategies.

*Virtual-torus embedding.* We decouple the basis frequency from the physical domain size by choosing $\omega_j < 2\pi/L_j$, which corresponds to a larger period $L_j' = 2\pi/\omega_j > L_j$. We represent $u$ by a LoRFS expansion defined on the larger periodic box $\Omega' = \prod_{j=1}^{d}[a_j', b_j']$ with $b_j' - a_j' = L_j'$, while enforcing the PDE and boundary/initial conditions only on $\Omega$. Since $\Omega$ is a strict subset of the period, values at the physical boundaries need not match.

*Windowed enforcement.* Alternatively, we keep a convenient periodic box $\Omega'$ (e.g., chosen so the Fourier modes are orthogonal on $\Omega'$) and restrict training to $\Omega$ by introducing a smooth window $w : \Omega' \to [0, 1]$ with $w \equiv 1$ on $\Omega$ and $w \approx 0$ outside. We optimize the weighted objective

$$\mathcal{J}(u) = \int_{\Omega'} w(x)^2 \, \|L[u](x)\|^2 \, dx \; + \; \lambda \int_{\partial\Omega} \|B[u](x)\|^2 \, dS(x),$$

where $w$ is chosen smooth to avoid spectral ringing and is represented by an auxiliary LoRFS model to preserve analytic integrability.

## B.5. Additional Experiments

**Navier–Stokes.** We further evaluate LoRFS in a challenging nonlinear setting by considering the two-dimensional Navier–Stokes equation in vorticity form. We use the same benchmark family as in FNO (Li et al., 2020), with viscosity $\nu = 10^{-4}$. The governing equation is

$$\partial_t \omega + u \cdot \nabla \omega = \nu \Delta \omega + f, \tag{41}$$

where the velocity field $u$ is recovered from the streamfunction $\psi$. Equivalently, we write the system as

$$\partial_t \omega + \psi_y \omega_x - \psi_x \omega_y - \nu \Delta \omega - f = 0, \tag{42}$$
$$\Delta \psi + \omega = 0. \tag{43}$$

For each trajectory, corresponding to a sampled initial condition from the FNO dataset, we optimize two coupled LoRFS models: one for the vorticity $\omega$ and one for the streamfunction $\psi$. The training objective contains three terms:

1. the vorticity-form Navier–Stokes residual,

2. the Poisson coupling residual $\Delta \psi + \omega = 0$,

3. an initial-condition matching term at $t = 0$.

In this implementation, the nonlinear transport term is handled analytically through LoRFS differentiation and multiplication. The only projected component is the initial condition.

Across five trajectories, LoRFS achieved

$$\text{rRMSE} = 0.07345 \pm 0.01547, \tag{44}$$

$$\text{rMAE} = 0.06811 \pm 0.01438. \tag{45}$$

These results provide encouraging evidence that LoRFS can be applied to nonlinear PDEs beyond the original benchmark suite.

