# OpenReview forum: "Overcoming PINNs Failure Modes In High Dimension With Low-Rank Fourier Sum"
_ICML.cc/2026/Conference — ICML 2026 spotlight_

### Official Review · Reviewer_1UeU · 2026-03-06

**Soundness:** 2
**Presentation:** 2
**Significance:** 3
**Originality:** 2
**Overall Recommendation:** 4
**Confidence:** 4

**Summary:**

This paper addresses the challenge of applying Physics-Informed Neural Networks (PINNs) to high-dimensional PDEs, where traditional PINN approaches often suffer from poor scalability and training instability due to the curse of dimensionality. To mitigate these limitations, the authors propose a novel solution representation based on low-rank Fourier sums (LoRFS), which leverages the spectral structure of high-dimensional functions to enable efficient and accurate approximation. The method is theoretically justified through analysis of its approximation properties and scalability in high dimensions. Empirical results on benchmark PDEs demonstrate that LoRFS-PINNs achieve superior accuracy and convergence compared to standard PINN baselines, particularly in problems with high input dimensionality.

**Compliance With Llm Reviewing Policy:**

Affirmed.

**Final Justification:**

I am satisfied that my concerns have been adequately addressed by the rebuttal, and I am willing to raise my score accordingly.

**Key Questions For Authors:**

1.	The paper identifies high-dimensional failure modes of PINNs as a central motivation, yet the root cause of this failure is not fully elaborated. Could you provide a more concrete characterization of the underlying mechanisms (such as curse of dimensionality in sampling, gradient sparsity, or loss imbalance) that lead to training instability or poor convergence? Specifically, how does this failure manifest in practice: does it result in convergence to incorrect or physically inconsistent solutions, or primarily in extremely slow convergence rates? Are there identifiable early warning signs during training (e.g., loss stagnation, gradient vanishing) that signal the onset of this failure mode?
2.	The proposed LoRFS framework is currently evaluated on forward problems. To what extent is it applicable to inverse problems (such as parameter estimation, coefficient learning, or system identification) where the solution depends on trainable parameters and additional data losses are common? Can the analytic computation of the physics loss be seamlessly integrated with data fidelity terms, regularization, or experimental noise models? Are there any theoretical or practical limitations (such as increased nonlinearity, ill-conditioning, or rank constraints) that could hinder performance in high-dimensional inverse settings?

**Limitations:**

The limitations have been addressed adequately, particularly in highlighting the method’s dependence on problem structure, and its scalability and applicability to non-periodic or non-sinusoidal problems. However, as raised in the key questions, the paper could further expand on the practical challenges in inverse settings, such as the integration of analytic loss computation with data-driven terms, potential ill-conditioning in high-dimensional parameter spaces, and the impact of rank constraints on model expressiveness. While these are not explicitly detailed in the current limitations, they represent important considerations for real-world deployment. Addressing them would strengthen the method’s perceived robustness and generalizability beyond the current experimental scope.

**Strengths And Weaknesses:**

$\textbf{Soundness}$
- The paper presents a technically sound approach to addressing high-dimensional PDEs via the Low-Rank Fourier Sum (LoRFS) representation, and the overall methodology is well-motivated. However, several critical gaps limit the strength of its technical foundation.
- The paper identifies the "failure mode" of PINNs in high dimensions as a central challenge, but this failure mode is not clearly defined or empirically substantiated. While the introduction (line 30) references the degradation of sampling-based approximations with increasing dimensionality, this claim lacks formal justification or supporting evidence. The paper does not analyze how or when PINNs fail in high dimensions, e.g., whether it manifests as optimization stagnation, poor convergence, or divergence, and no experiments are designed to diagnose these failure modes explicitly. This weakens the motivation for the proposed method.
- The theoretical analysis is a strong point, yet it remains disconnected from the empirical validation. For instance, if the theory claims dimension-independent approximation error convergence, this should be demonstrated empirically by measuring error scaling across increasing dimensions. While Table 1 (low-dimension) and Table 2 (high-dimension) provide some comparative results, they do not systematically assess performance as a function of dimensionality. A more rigorous analysis, such as error vs. dimension curves, would significantly strengthen the scalability claim.
- The choice of benchmarks is noteworthy: many test problems exhibit periodic boundary conditions and solutions that are naturally represented in trigonometric bases (e.g., Fourier modes), which inherently favor the LoRFS approach. This raises concerns about overfitting the evaluation to a specific class of problems. To support broader generalizability, additional experiments on non-periodic or non-sinusoidal problems, such as those with sharp gradients, discontinuities, or non-oscillatory behavior, would be valuable. The reaction system is an exception, but more diverse test cases are needed to assess robustness.

$\textbf{Presentation}$
- The paper is generally well-structured, progressing logically from problem formulation to theoretical analysis and empirical validation. The narrative is clear and accessible.
- However, the theoretical and empirical components are not sufficiently integrated. For example, if theorems establish that LoRFS enables dimension-independent approximation, this should be corroborated with empirical plots showing error scaling that aligns with the theoretical prediction. Without such cross-validation, the theoretical claims remain under-supported.
- The method significantly alters the standard PINN architecture, yet the paper lacks a visual or detailed description of the network’s solution representation. A schematic diagram of the LoRFS-based architecture (showing how the Fourier coefficients are parameterized, learned, and enforced via the physics loss) would greatly improve clarity.
- While the related work section is comprehensive, it omits key connections to the growing body of spectral and frequency-based methods in PINNs. Works such as [Wong et al., 2022] on sinusoidal activation and spectral learning, [Xia et al., 2023] and [Yu et al., 2025] on spectral PINNs are highly relevant and demonstrate prior art in leveraging trigonometric representations. The authors should explicitly position their work within this context, clarifying how LoRFS differs in principle, design, or effectiveness from these approaches.

$\textbf{Significance}$
- The paper addresses a critical and timely challenge: the scalability of PINNs to high-dimensional PDEs, a major bottleneck in real-world scientific computing. Successfully tackling this issue has the potential to significantly expand the applicability of PINNs in fields such as fluid dynamics, quantum mechanics, and uncertainty quantification.
- However, the significance is somewhat tempered by the lack of systematic scalability analysis and limited benchmark diversity. Without clear evidence that the method generalizes beyond trigonometric or periodic problems, its broader impact remains uncertain. A more rigorous evaluation across a wider range of PDE types and boundary conditions would elevate its significance.

$\textbf{Originality}$
- The core idea of representing solutions via low-rank Fourier sums is conceptually aligned with recent advances in spectral learning for neural networks. While the specific implementation and integration into PINNs may offer practical benefits, the novelty is not clearly articulated.
- The paper does not sufficiently distinguish itself from existing spectral methods in PINNs, particularly those that also exploit trigonometric or frequency-based representations. The limited discussion of related work makes it difficult to assess whether LoRFS introduces a new principle, a novel architecture, or merely a variant of prior approaches. To strengthen the claim of originality, the authors should explicitly contrast their method with existing spectral PINN frameworks, highlighting unique advantages in approximation efficiency, generalization, or computational cost.

$\textbf{References}$

[Wong et al., 2022] Wong, Jian Cheng, et al. "Learning in sinusoidal spaces with physics-informed neural networks." IEEE Transactions on Artificial Intelligence 5.3 (2022): 985-1000.

[Xia et al., 2023] Xia, Mingtao, Lucas Böttcher, and Tom Chou. "Spectrally adapted physics-informed neural networks for solving unbounded domain problems." Machine Learning: Science and Technology 4.2 (2023): 025024.

[Yu et al., 2025] Yu, Tianchi, et al. "Spectral informed neural networks." Journal of Computational and Applied Mathematics (2025): 117178.

---

> ### Author Rebuttal · Authors · 2026-03-30
>
> We thank the reviewer for the detailed comments.
>
> > “The paper identifies the ‘failure mode’ of PINNs in high dimensions as a central challenge, but this failure mode is not clearly defined or empirically substantiated...”
>
> > “The paper identifies high-dimensional failure modes of PINNs as a central motivation, yet the root cause of this failure is not fully elaborated...”
>
> By “failure mode” we mean the well-documented [1,2,3] regime in which PINNs either (i) converge to an incorrect, often overly smooth, solution despite decreasing residual loss, or (ii) stagnate and fail to recover the target dynamics within practical budgets. The two mechanisms emphasized in our paper are spectral bias toward low frequencies and degradation of collocation-based residual estimation in high dimensions due to numerical integration error and gradient variance. Fig. 1 illustrates this directly: the training loss decreases while the relative solution error remains high.
>
> > “The theoretical analysis is a strong point, yet it remains disconnected from the empirical validation...”
>
> The theory and experiments play complementary roles. Theorems 4.2–4.7 show that LoRFS avoids tensor-product growth and admits favorable high-dimensional approximation scaling for broad smooth function classes, while the experiments test whether these advantages translate into practice on canonical PINN failure-mode benchmarks and their high-dimensional extensions. This connection is reflected in Table 2, where strong PINN baselines degrade sharply with dimension while LoRFS remains accurate and stable.
>
> > “... many test problems exhibit periodic boundary conditions and solutions that are naturally represented in trigonometric bases. ... Without clear evidence that the method generalizes beyond trigonometric or periodic problems, its broader impact remains uncertain.”
>
> These are not hand picked problems, but standard failure-mode benchmarks from the PINN literature [3,4]. Moreover, the reaction benchmark does not trivially align with the LoRFS basis, yet LoRFS still performs strongly. Our time-dependent benchmarks also treat $t$ as a non-periodic coordinate, and the wave benchmark itself does not impose periodic boundary conditions, Section 4.3 further explains how LoRFS can handle non-periodic settings. We also include a Navier–Stokes experiment in which neither the initial condition nor the solution is given in closed form, and therefore neither is explicitly aligned with the basis, see our response to Reviewer 7Asa for details. Taken together, these results show that the evaluation is not restricted to simple closed-form trigonometric targets.
>
> > “The method significantly alters the standard PINN architecture, yet the paper lacks a visual or detailed description of the network's solution representation...”
>
> > “...it omits key connections to spectral and frequency-based PINN methods...”
>
> > “The novelty is not clearly articulated.”
>
> > “The paper does not sufficiently distinguish itself from existing spectral methods in PINNs...”
>
> LoRFS is not a modified neural-network architecture, but a low-rank separable Fourier representation
> $g_\theta(x)=\sum_{r=1}^{s}\prod_{j=1}^{d} f_{rj}(x_j;\theta_{rj}),$
> where each factor is a truncated 1D Fourier series with learnable Fourier coefficients. This is a key distinction from the existing spectral methods in PINNs. [5] remain within the standard sampled-residual PINN paradigm and mainly modify the input representation. [6,7] also change the solution representation by learning spectral coefficients rather than the physical solution directly. However, these methods still optimize a sampled residual objective. In contrast, LoRFS combines a different solution representation with, for tractable PDE terms, analytic continuous-domain evaluation of the physics loss via factorized integrals rather than sampled collocation.
>
> > “To what extent is it applicable to inverse problems?”
>
> LoRFS is structurally compatible with inverse problems, the analytic physics loss is one differentiable objective term and can be combined with data-fidelity, regularization, and noise-aware likelihood terms. However, inverse problems are outside this paper’s scope, so we leave them to future work.
>
> [1] Krishnapriyan, et al. "Characterizing possible failure modes in physics-informed neural networks" (2021)
>
> [2] Rathore, P. et al. "Challenges in training pinns: A loss landscape perspective" (2024)
>
> [3] Xu, C., et al. "Fp64 is all you need: Rethinking failure modes in physics-informed neural networks" (2025)
>
> [4] Wu, H., et al. "Ropinn: Region optimized physics-informed neural networks." (2024)
>
> [5] Wong, Jian Cheng, et al. "Learning in sinusoidal spaces with physics-informed neural networks." (2022)
>
> [6] Xia, Mingtao, et al. "Spectrally adapted physics-informed neural networks for solving unbounded domain problems." (2023)
>
> [7] Yu, Tianchi, et al. "Spectral informed neural networks." (2025)

---

> > ### Author Rebuttal · Reviewer_1UeU · 2026-04-02
> >
> > Thank you to the authors for their detailed and thorough rebuttal. I am satisfied that my concerns have been adequately addressed, and I am willing to raise my score accordingly.
> >
> > I note that some remaining concerns regarding reproducibility should be resolved by the authors' commitment to releasing the code upon acceptance, which I welcome.
> >
> > Overall, I believe this is a promising contribution and look forward to seeing the final version of the work.

---

### Official Review · Reviewer_DQbr · 2026-03-11

**Soundness:** 3
**Presentation:** 3
**Significance:** 3
**Originality:** 3
**Overall Recommendation:** 4
**Confidence:** 3

**Summary:**

The paper proposes Low-Rank Fourier Sums, representing the PDE solution as a sum of rank-one separable Fourier factors across coordinates, with the goal of making oscillatory structure explicit and enabling closed-form evaluation of common residual and variational losses. This work also gives approximation-theoretic results for continuous, Sobolev, mixed-Sobolev, and analytic function classes, and reports strong empirical performance on canonical PINN failure-mode benchmarks, their high-dimensional extensions, and a biharmonic PDE. A relevant problem considered here is that standard PINNs optimize a Monte Carlo approximation of a continuous-domain residual objective, which can be noisy and unstable in high dimensions. The paper’s central claim is that LoRFS avoids this by using a separable Fourier parameterization whose derivatives and integrals can often be computed analytically.

**Compliance With Llm Reviewing Policy:**

Affirmed.

**Final Justification:**

My concerns and questions have been fully addressed or appropriately acknowledged.

This does not change my rating, however, because the limitations acknowledged by the authors still constrain the overall impact of the work. My original score (4) already reflected this position.

**Key Questions For Authors:**

1. What happens when the target solution is not low-rank separable? How sensitive is performance to the chosen rank $s$ on harder, less structured problems?
2. What are the training costs of LoRFS compared with the baselines?
3. Are there cases where LoRFS converges to a low loss but still gives a poor solution, similar to standard PINN failure modes?
4. How does the method perform on PDEs with less regular, non-periodic, or localized solutions?

**Limitations:**

Limitations are not explicitly mentioned. However, I do not think that are any limitations that must be mentioned.

**Strengths And Weaknesses:**

# Strenghs
1. The method is simple and well matched to oscillatory PDEs.
2. Fourier factors are a natural fit for wave-like, multiscale, and stiff problems where vanilla PINNs often fail.
3. The low-rank structure is meaningful: compared with a full tensor Fourier basis, the parameterization is much cheaper when the target is approximately separable.

# Weaknesses
1. The paper strongly suggests that LoRFS removes collocation noise entirely. But for non-spectral terms, it introduces an auxiliary regression / projection stage that is trained approximately. So the “no sampling” story is not universally true.
2. The approach will work best when the PDE solution is well approximated by a small number of separable Fourier terms. So, it somewhat limits the scope of this work. The paper does not deeply investigate cases where this assumption fails.
3. The paper says it replaces the collocation paradigm with exact closed-form losses. That is too strong if some benchmarks still rely on an auxiliary approximated projection stage. I feel this is an over-statement.

---

> ### Author Rebuttal · Authors · 2026-03-30
>
> We thank the reviewer for the helpful and balanced feedback.
>
> > “The paper strongly suggests that LoRFS removes collocation noise entirely. But for non-spectral terms, it introduces an auxiliary regression / projection stage that is trained approximately. So the ‘no sampling’ story is not universally true.”
>
> We agree that this wording is too broad. What we mean is the following: when all PDE terms are analytically tractable in the LoRFS algebra, the residual objective can indeed be evaluated analytically, without collocation-based estimation. However, for non-spectral terms handled through the auxiliary projection step, approximation error is reintroduced through that regression stage. We will clarify this explicitly and include it in the limitations section of the final version.
>
> > “The approach will work best when the PDE solution is well approximated by a small number of separable Fourier terms...”
>
> LoRFS is most attractive when the target admits a reasonably compact low-rank Fourier representation. If the solution is not well compressed in this form, the required rank $s$ and possibly the frequency cutoff $K$ must increase, and the efficiency advantage can diminish. We will make this limitation more explicit in the final version. To partially broaden the empirical picture, we also added a limited Navier–Stokes experiment in a challenging nonlinear regime (see our response to Reviewer 7Asa) and got favorable results. This case is also meaningful because neither the initial condition nor the solution is given in closed form, unlike the original benchmarks.
>
> > “The paper says it replaces the collocation paradigm with exact closed-form losses. That is too strong if some benchmarks still rely on an auxiliary approximated projection stage...”
>
> We agree and will soften this claim. A more accurate statement is that LoRFS replaces collocation-based residual estimation with analytic loss evaluation when the PDE terms are closed under the LoRFS operations. When auxiliary projection is used, the final optimization still benefits from analytic loss evaluation, but the projected component itself is only approximate. We will revise the wording accordingly and state this explicitly in the limitations section of the final version.
>
> > “What happens when the target solution is not low-rank separable? How sensitive is performance to the chosen rank on harder, less structured problems?”
>
> If the target is not well approximated by a low-rank separable Fourier structure, then LoRFS will require larger $s$ (and in some cases larger $K$) to achieve good accuracy. In that sense, the method is sensitive to underestimating rank on harder or less structured problems: if $s$ is too small, one should expect underfitting. More broadly, our theoretical message is not that every function is efficiently representable, but rather that for important smooth function classes the complexity can be dominated by $s$ without tensor-product growth in $d$. We will clarify this scope more directly.
>
> > “What are the training costs of LoRFS compared with the baselines?”
>
> For the high-dimensional comparison in Table 2, we already used a controlled one-hour budget on the same hardware across methods. In addition, to address the same concern on the standard 2D benchmarks, we reran PINN FP64 and PINNMamba under the same controlled setup, please refer to our response to Reviewer 7Asa for details.
>
> > “Are there cases where LoRFS converges to a low loss but still gives a poor solution, similar to standard PINN failure modes?”
>
> Yes, this can happen if an auxiliary projected term is inaccurate. However, on the benchmarks we reported, we did not observe the same pronounced “low training loss but clearly wrong solution” pathology that standard PINNs exhibit in Fig. 1. Our interpretation is not that such behavior becomes impossible, but that LoRFS removes a major source of it by optimizing the continuous residual objective analytically rather than through noisy collocation estimates.
>
> > “How does the method perform on PDEs with less regular, non-periodic, or localized solutions?”
>
> For less regular solutions, especially those with true discontinuities, we expect LoRFS to perform worse because Fourier representations exhibit Gibbs-type artifacts near jumps. Our framework can handle non-periodic problems through the mechanisms described in Section 4.3. For localized yet smooth solutions, LoRFS can still be applicable, but highly concentrated structure may require larger $s$ and $K$.

---

> > ### Author Rebuttal · Reviewer_DQbr · 2026-04-02
> >
> > Thanks to the authors for their response and for being upfront about certain limitations. My concerns and questions have been fully addressed or appropriately acknowledged. I would therefore like to maintain my score of 4.

---

### Official Review · Reviewer_7Asa · 2026-03-12

**Soundness:** 3
**Presentation:** 3
**Significance:** 3
**Originality:** 3
**Overall Recommendation:** 5
**Confidence:** 4

**Summary:**

this paper proposes Low-Rank Fourier Sums (LoRFS), a method for solving PDEs that sidesteps the two main failure modes of PINNs, sampling noise and spectral bias, by representing the solution as a sum of rank-1 separable Fourier components:
$$u(x) = \sum_{r=1}^{s} \prod_{j=1}^{d} f_{rj}(x_j; \theta_{rj})$$
where each $f_{rj}$ is a truncated 1D Fourier series. Because the domain $\Omega$ factorizes into $\Omega_1 \times \cdots \times \Omega_d$, integrals over the domain reduce to products of 1D integrals, each of which has a closed-form solution. this means the Galerkin residual, which standard PINNs approximate via Monte Carlo, can be computed exactly, removing both the sampling variance and the spectral bias at once. The paper proves approximation bounds for three function classes (Sobolev $H^m$, mixed Sobolev $H^m_{\text{mix}}$, Gevrey), tests on Convection, Reaction, Wave, and Biharmonic up to $d=9$, and achieves dramatic accuracy improvements over PINN baselines with only 9k parameters.

**Compliance With Llm Reviewing Policy:**

Affirmed.

**Final Justification:**

My concerns have been largely addressed. The Navier-Stokes experiment provides meaningful evidence that the nonlinear term does not require projection, and the turbulence limitation is now explicitly acknowledged. The controlled-budget replication clarifies the Table 1 comparison. The NS experiment is at modest Reynolds number rather than a turbulent regime, so the contribution to chaotic settings remains an open question, but the authors are honest about this boundary.

**Key Questions For Authors:**

1. The Fourier separable structure allows analytic computation for linear operators and polynomial nonlinearities. For Navier-Stokes or other strongly nonlinear PDEs, would the nonlinear term $\mathbf{u} \cdot \nabla \mathbf{u}$ still admit a closed-form integral over the product basis, or would projection (Section 4.3) always be required? If projection is needed, how does the projection error scale with turbulence intensity or Reynolds number? This is arguably the most important PINN failure mode in practice, and understanding whether LoRFS can make progress there would substantially affect the significance assessment.

2. Classical Fourier-Galerkin spectral methods (e.g., DEDALUS) also evaluate residuals analytically for smooth periodic problems and avoid sampling noise entirely. For the benchmarks in Table 1, how does LoRFS compare to a tuned pseudospectral implementation? Even a qualitative discussion of where LoRFS has advantages over classical codes, for example, parameterization flexibility or compatibility with data-driven approaches, would help readers from the scientific computing community understand the contribution's positioning.

3. Table 1 baselines come from Xu et al. (2025b) with different hardware and training setups, while LoRFS results are from the authors' implementation. could the authors replicate even two of the strongest Table 1 baselines (PINN FP64 and PINNMamba) under the same one-hour L4 GPU budget used for Table 2? This would make the standard-dim comparison as controlled as the high-dim one.

4. for chaotic or turbulent PDEs where energy is distributed across all wavenumbers, the rank $s$ needed for accurate approximation would presumably grow, possibly exponentially. Is there a regime (characterized by, say, a Kolmogorov length scale or spectral decay rate) where LoRFS is expected to fail or become computationally intractable? Understanding this boundary would help readers assess the scope of applicability more precisely than "future work" framing allows.

5. the Biharmonic experiment (Table 3) goes to $d=10$. How much does the parameter count grow from $d=9$ to $d=10$ with $K=30, s=30$, and is there a practical dimensionality ceiling before the $O(d)$ parameter scaling advantage breaks down?

**Limitations:**

partially addressed. the paper explicitly discusses nonlinear PDE limitations and non-periodic boundary condition handling in Appendix B.4. What is missing is an honest acknowledgment that the structural incompatibility with chaotic or turbulent dynamics is a harder barrier than "future work" framing implies, it is not simply that the experiments were not run, but that the low-rank Fourier structure is architecturally mismatched with turbulent solutions. the missing classical spectral comparison is also not flagged as a limitation.

**Strengths And Weaknesses:**

This is a genuinely strong paper. The core insight is elegant in the way that good theoretical ML contributions tend to be, it takes a specific, well-understood problem (why do PINNs fail on oscillatory high-dimensional PDEs?), identifies two precise mechanisms (sampling noise and spectral bias), and shows that a single architectural change eliminates both simultaneously. That's a clean story and it's backed up by the mathematics.

the theory is the strongest part. Theorems 4.2--4.7 give approximation bounds for three different function classes with convergence rates that don't degrade with dimension $d$, the bottleneck is always the rank $s$ and the smoothness of the target function, not $d$. Theorem 4.7 in particular gives exponential convergence $\varepsilon \leq \|u\|_{G_{\sigma,0}} \cdot d \cdot e^{-2\min_i(\sigma_i)K}$ for Gevrey-regular functions, which is exactly the regime (analytic solutions, smooth coefficients) where many physical problems of interest live. the proofs in Appendix A hold up under checking, Lemma A.6 and the reduction to Theorems A.8--A.10 via Theorem A.5 are handled correctly.

the empirical results are dramatic and, based on what is reported, fairly presented. On standard-dim convection, LoRFS achieves rRMSE $9 \times 10^{-6}$ versus PINN FP64 at $0.0072$, roughly $800\times$ better against the strongest baseline, this isn't a cherry-picked comparison: PINNMamba actually performs worse than PINN FP64 on convection (rRMSE $0.0197$ vs. $0.0072$), so the $800\times$ figure is conservative, not optimistic. Table 2 high-dimensional results are also genuinely impressive, at $d=9$ on convection, PINNMamba gives rRMSE $> 1.0$ (essentially random) while LoRFS stays at $0.006$. Parameter efficiency is real too: 9k parameters for LoRFS versus 285k for PINNMamba and 33k for PINN FP64.

the primary concern is that the scope of the paper's claims may be overstated by the title. All four benchmarks are either linear (Convection, Wave, Biharmonic) or have a mild nonlinearity with a known analytic solution (Reaction, which is logistic growth). No chaotic systems, no turbulence, and nothing that tests whether the method generalizes to PDEs where the solution lacks a compact low-rank Fourier structure. This isn't simply a matter of not having run additional experiments, there's a structural reason for the gap. For turbulent flows, energy is distributed across all wavenumbers following the Kolmogorov cascade, so a low-rank Fourier decomposition can't represent the solution compactly regardless of how large $K$ and $s$ are. The title "Overcoming PINNs Failure Modes" sets an expectation of broad coverage that the method can't fulfill for this important class of problems. The paper positions the limitation as future work, but it's more accurately an architectural constraint.

a missing comparison is with classical spectral Galerkin methods. DEDALUS and similar pseudospectral codes also evaluate residuals analytically for smooth periodic problems and avoid sampling noise entirely, they address the same two failure modes that motivate LoRFS. Readers from physics and applied mathematics communities will naturally ask why LoRFS is preferable to classical spectral methods on these benchmarks. Even a qualitative discussion of where LoRFS has advantages (e.g., parameterization flexibility, automatic differentiation, integration with data-driven priors) would help contextualize the contribution.

one additional experimental concern: Table 1 baseline numbers are taken from Xu et al. (2025b) rather than from the authors' own replication runs. Only Table 2 uses a controlled one-hour budget on identical hardware. This makes Table 1 a mixed comparison, even if the authors acknowledge this honestly.

---

> ### Author Rebuttal · Authors · 2026-03-30
>
> We thank the reviewer for the careful reading and constructive feedback.
>
> > “For Navier–Stokes or other strongly nonlinear PDEs, would the nonlinear term still admit a closed-form integral over the product basis, or would projection be required? If projection is needed, how does the projection error scale with turbulence intensity or Reynolds number?”
>
> In our setting, the nonlinear term does not need to be projected. To address this, we ran a limited additional experiment on the 2D Navier–Stokes equation in vorticity form using the same benchmark family as in FNO [1], with viscosity $\nu = 10^{-4}$. The PDE is
> $\partial_t \omega + u \cdot \nabla \omega = \nu \Delta \omega + f,$
> with $u$ recovered from the streamfunction $\psi$, equivalently
> $\partial_t \omega + \psi_y \omega_x - \psi_x \omega_y - \nu \Delta \omega - f = 0,$
> and
> $\Delta \psi + \omega = 0.$
>
> We solve each trajectory (a sampled initial condition from the FNO dataset) with LoRFS by optimizing two coupled LoRFS models, one for $\omega$ and one for $\psi$. The loss includes: (i) the vorticity-form Navier–Stokes residual, (ii) the Poisson coupling residual $\Delta \psi + \omega = 0$, and (iii) the initial-condition matching term at $t=0$. In this implementation, the nonlinear transport term is handled analytically through LoRFS differentiation and multiplication; the only projected component is the initial condition. On 5 trajectories, we obtained rRMSE $= 0.07345 \pm 0.01547$ and rMAE $= 0.06811 \pm 0.01438$.
>
> We view this as encouraging evidence that LoRFS can tackle nonlinear PDEs beyond the original benchmark suite.
>
> > “How should LoRFS be positioned relative to classical Fourier-Galerkin / pseudospectral methods, which also avoid sampling noise on smooth periodic problems?”
>
> We agree that this positioning should be clarified. We do not expect LoRFS to outperform tuned pseudospectral solvers in low-dimensional settings; classical spectral methods are likely stronger in both raw accuracy and speed there. Our intended point of comparison is different. First, LoRFS replaces the full tensor-product spectral coefficient grid with an explicit low-rank separable continuous representation, which is specifically aimed at improving scalability in higher dimensions. Second, LoRFS remains within the same optimization-based framework as PINN-style methods, making it straightforward to combine analytic physics losses with data terms, inverse problems, and partially known physics. We will add this qualitative positioning explicitly, while also clarifying that classical spectral solvers remain a very strong baseline in low-dimensional smooth regimes.
>
> > “Could the authors rerun the strongest Table 1 baselines under the same one-hour L4 GPU budget used for Table 2?”
>
> We ran PINN FP64 and PINNMamba on the standard 2D benchmarks under the same settings as in our paper. For convection, PINNMamba obtained rRMSE $0.889$ and rMAE $0.860$, while PINN FP64 obtained rRMSE $1.37$ and rMAE $1.23$. For wave, PINNMamba obtained rRMSE $0.523$ and rMAE $0.512$, while PINN FP64 obtained rRMSE $0.152$ and rMAE $0.146$. For reaction, PINNMamba obtained rRMSE $0.0137$ and rMAE $0.0062$, while PINN FP64 obtained rRMSE $0.769$ and rMAE $0.762$.
>
> We suspect the discrepancy relative to the originally reported results is largely due to compute budget. The original works do not explicitly specify their training budget, whereas in our reruns we used their released implementations but terminated training after 60 minutes on a single L4 GPU. In all cases except the reaction benchmark for PINNMamba, training was stopped mid-run by this budget constraint.
>
> > “For chaotic or turbulent PDEs, is there a regime where the required rank becomes so large that LoRFS loses its practical advantage?”
>
> Yes. We expect LoRFS to become less attractive when the solution is no longer reasonably compressible in a low-rank Fourier form, or when energy remains spread across many wavenumbers so that modest $s, K$ truncations are ineffective. In such regimes, both $s$ and $K$ may need to grow substantially, reducing the practical advantage of LoRFS. We do not currently claim a sharp boundary in terms of a Kolmogorov scale or spectral-decay threshold, and we will state this limitation explicitly in the final version.
>
> > “In the biharmonic experiment, how much does the parameter count increase from $d=9$ to $d=10$ for $K=30, s=30$, and is there a practical dimensional ceiling?”
>
> For fixed rank and degree, the LoRFS parameter count scales as $s d (2K+1)$. Thus, with $s=30$ and $K=30$, increasing from $d=9$ to $d=10$ adds $1830$ parameters. More broadly, the practical ceiling is not due to the explicit linear dependence on $d$, but to whether the target problem remains accurately approximable with moderate rank and degree as dimension and complexity grow.
>
> [1] Li, Zongyi, et al. "Fourier neural operator for parametric partial differential equations." arXiv preprint arXiv:2010.08895 (2020).

---

> > ### Author Rebuttal · Reviewer_7Asa · 2026-04-01
> >
> > All concerns have been addressed. The Navier-Stokes experiment and the honest acknowledgment of the turbulence limitation address the primary concern regarding scope. The controlled-budget replication clarifies the comparison in Table 1. No further questions.

---

### Official Review · Reviewer_464s · 2026-03-12

**Soundness:** 3
**Presentation:** 3
**Significance:** 3
**Originality:** 3
**Overall Recommendation:** 4
**Confidence:** 3

**Summary:**

The authors propose Low-Rank Fourier Sums (LoRFS), replacing the coordinate-wise neural networks typically used in PINNs with a separable low-rank sum of 1D complex Fourier series. This architecture converts the standard Monte-Carlo estimation of the residual loss into an exact, analytic closed-form integration. Furthermore, it replaces the automatic differentiation (AD) of the PDE operators with exact spectral differentiation.

By avoiding collocation point sampling and AD, and by natively employing a spectral basis, LoRFS mitigates spectral bias and stabilizes the optimization landscape for high-order and high-dimensional PDEs. Theoretically, the authors demonstrate that for sufficiently smooth functions, the parameter count scales roughly linearly with dimension, effectively mitigating the curse of dimensionality inherent to standard spectral methods. For practical implementation on non-linear or non-spectral PDE terms, however, the method falls back to an approximated regression-based projection strategy using auxiliary models.

**Compliance With Llm Reviewing Policy:**

Affirmed.

**Final Justification:**

My concerns are mostly addressed. I am keeping my overall rating at 4, however, as it is still not clear exactly how the model scales with long time horizons, and where and when artifacts like Gibbs phenomena may appear. While acknowledged by the authors, additional experiments studying such cases are not provided.

**Key Questions For Authors:**

1. How would the rank $s$ scale for the complex dynamics pointed out in the Weaknesses?
2. When solving PDEs with discontinuities in their solution (e.g. a 1D Burgers' equation), how would LoRFS behave, especially considering that the method relies on analytical spectral differentiation?
3. I am also interested in knowing whether the temporal horizon of the PDE solution affects the rank $s$ and the number of frequencies $K$, especially with the more complex dynamics. The authors pointed out PINNs' challenges with solving for long-time dynamics, but I did not notice any experiments addressing particularly long time-horizons.

**Limitations:**

yes

**Strengths And Weaknesses:**

## Strengths
- The paper presents a well-motivated, original method backed by strong theoretical results. It effectively tackles multiple recognized challenges in standard PINNs simultaneously.

- The authors demonstrate that LoRFS is remarkably stable for the 4th-order biharmonic equation, that its parameter count scales linearly with the number of dimensions, and that spectral bias is effectively mitigated by natively employing a spectral representation.

- The paper is well-written and easy to follow. The motivations are clearly described in the introduction, the methodology is accessible, and the theoretical results follow a logical progression.

## Weaknesses
My primary concern is whether LoRFS can serve as a viable replacement for typical neural-network-based PINNs across a wider range of PDEs. While, in fairness to LoRFS, standard PINNs also struggle with the dynamics discussed below, the strict spectral representation may impose unique restrictions in these regimes:

- **Complex non-linear dynamics:** The authors showed that LoRFS excels in solving linear and smoothly *oscillating* PDEs. However, its scalability to complex multi-scale non-linear dynamics, such as vortex shedding and turbulence in Navier-Stokes equations, as well as chaotic regimes in the Kuramoto-Sivashinsky equation, is highly questionable.

- **Discontinuities and shock waves:** Similarly, PDEs that do not conform to the smoothness requirements, such as 1D Burgers' equation, will inevitably introduce spectral artifacts like the Gibbs phenomenon.

- **Complex geometries:** The proposed heuristic for handling non-periodic boundaries is restrictive for modelling more complex geometries (e.g. modelling the flow over a bluff body).

---

> ### Author Rebuttal · Authors · 2026-03-30
>
> We thank the reviewer for the thoughtful and constructive feedback.
>
> > “My primary concern is whether LoRFS can serve as a viable replacement for typical neural-network-based PINNs across a wider range of PDEs...”
> > - “Discontinuities and shock waves: ...”
> > - “Complex geometries: ...”
> > - “Complex non-linear dynamics: ...”
>
> **Discontinuities and shock waves.**
> We agree. LoRFS is based on a continuous Fourier representation, and our theoretical results also assume smoothness / regularity. For solutions with discontinuities or shocks, we expect Gibbs-type artifacts and therefore weaker performance. In the final version, we will add a limitations paragraph stating this explicitly.
>
> **Complex geometries.**
> LoRFS is indeed restricted to rectangular / hypercube-like domains. In some simple cases, this can be partially alleviated by a change of variables that maps the problem to a box-like coordinate system. For example, one may use polar / radial coordinates for disks or spheres, after which the same general techniques described in the paper can be applied in the transformed coordinates. We will clarify this limitation more directly in the revision.
>
> **Complex non-linear dynamics.**
> We agree that these regimes require further testing. LoRFS can still be applied in such settings, but whether it remains efficient depends on whether the solution admits a reasonably compact low-rank Fourier representation. To partially probe this, we ran an additional Navier–Stokes experiment with the same settings as FNO [1] and obtained favorable results; more details are provided in our response to Reviewer 7Asa. More broadly, we agree that this direction deserves further examination, and we will clarify this limitation in the final version.
>
> > “How would the rank scale for the complex dynamics pointed out in the Weaknesses?”
>
> Our expectation is that as the dynamics become more complex, less smooth, and less compressible in a low-rank Fourier form, the required rank $s$ will increase, potentially substantially. Likewise, the needed frequency cutoff $K$ may also grow as more broadband structure must be represented.
>
> > “When solving PDEs with discontinuities in their solution (e.g., a 1D Burgers’ equation), how would LoRFS behave, especially considering that the method relies on analytical spectral differentiation?”
>
> We expect LoRFS to perform poorly in the presence of true discontinuities, precisely because spectral differentiation acts on a Fourier representation that will exhibit Gibbs-type oscillations near jumps.
>
> > “I am also interested in knowing whether the temporal horizon of the PDE solution affects the rank and the number of frequencies, especially with the more complex dynamics. The authors pointed out PINNs’ challenges with solving for long-time dynamics, but I did not notice any experiments addressing particularly long time-horizons.”
>
> Since time is treated as an additional coordinate in LoRFS, increasing the temporal horizon can require richer temporal frequency content and, in more complex regimes, potentially larger rank as well. In other words, for longer and more intricate dynamics, we expect both $K$ and $s$ may need to increase.
>
> [1] Li, Zongyi, et al. "Fourier neural operator for parametric partial differential equations." arXiv preprint arXiv:2010.08895 (2020).

---

> > ### Author Rebuttal · Reviewer_464s · 2026-04-04
> >
> > Thanks to the authors for their response and clarifications. Taking the additional NS experiment into account, my concerns are mostly addressed.

---

### Decision · Program_Chairs · 2026-04-30

**Decision:**

Accept (spotlight)

**Comment:**

The proposed Low-Rank Fourier Sums (LoRFS) method provides a mathematically elegant solution to standard Physics-Informed Neural Network (PINN) failure modes by replacing sampling-based collocation with exact, closed-form integration. Its primary strengths lie in its theoretical rigor, which demonstrates that parameter counts scale linearly with dimension ($O(d)$), and in its empirical dominance over standard baselines in high-dimensional, oscillatory, and high-order PDEs. However, the architecture faces significant structural weaknesses. It is natively restricted to rectangular domains and smooth, periodic solutions. Reviewers also highlighted its inevitable struggle with discontinuities (Gibbs phenomenon) and with chaotic, multiscale dynamics such as turbulence, where the low-rank assumption breaks down. While the authors provided promising Navier-Stokes results during rebuttal, the method remains a specialized tool for smooth regimes rather than a universal PINN replacement.